# UniOMA: Unified Optimal-Transport Multi-Modal Structural Alignment for Robot Perception

## Abstract

Achieving generalizable and well-aligned multimodal representation remains a core challenge in artificial intelligence. While recent approaches have attempted to align modalities by modeling conditional or higher-order statistical dependencies, they often fail to capture the structural coherence across modalities. In this work, we propose a novel multimodal alignment method that augments existing contrastive losses with a geometry-aware Gromov-Wasserstein (GW) distance-based regularization. To this end, we encode intra-modality geometry with modality-specific similarity matrices and extend the GW distance to minimize their discrepancies from a dynamically learned barycenter, thereby enforcing structural alignment across modalities beyond what is captured by InfoNCE-like mutual information objectives. We apply this optimal-transport-based alignment strategy to robot perception tasks involving underexplored modalities such as force and tactile signals, where modality data often exhibit varying sample densities. Experimental results show that our method yields superior inter-modal coherence and significantly improves downstream robot perception tasks such as robot and environment state prediction. Moreover, our GW-based augmentation term is versatile and can be seamlessly integrated into most InfoNCE-like objectives.

## 1 Introduction

The integration of information from diverse sources or modalities has received increasing attention across a wide range of AI applications, including image/video/text generation (Rombach et al., 2022; Mirza & Osindero, 2014), healthcare (Acosta et al., 2022), autonomous systems (Feng et al., 2021), and scientific discovery (Steyaert et al., 2023). Recent advances in contrastive self-supervised learning (CSSL) (He et al., 2020; Chen et al., 2020; Grill et al., 2020; Chen & He, 2021), particularly those leveraging InfoNCE losses (Oord et al., 2018), have shown strong performance in aligning heterogeneous modalities into a shared representation space (Radford et al., 2021). Such alignment has enabled zero-shot cross-modal retrieval, transfer, generation, and completion (Radford et al., 2021; Girdhar et al., 2023; Chen et al., 2023; Zhu et al., 2023; Luo et al., 2022). By maximizing agreement between paired modalities of the same instance while minimizing similarity between distinct instances, CSSL encourages the learning of invariant and semantically meaningful features.

While effective, InfoNCE-style objectives operate as binary classification losses that only discriminate positives from negatives (Wang & Isola, 2020), without explicitly modeling the continuous pairwise distance geometry within each modality. In multimodal alignment, this limitation produces what we call a *structural alignment gap* (Liang et al., 2022): embeddings may appear statistically aligned across modalities yet fail to preserve their intrinsic structural topologies. Our key insight is that multimodal alignment should not be limited to maximizing *population-level* statistical dependence between distributions of modality representations. It must also preserve *instance-level* geometric relationships within each modality. In other words, if $x_i$ is close to $x_j$, then their counterparts $y_i$ and $y_j$ should also remain close. Classic InfoNCE objectives, which are essentially a lower bound of Shannon's mutual information (Kraskov et al., 2004; Poole et al., 2019), rely on binary discrimination between positive and negative pairs. While effective at capturing population-level dependence, this approach is theoretically incapable of preserving intra-modal geometry, often leading to representations that are statistically aligned but structurally inconsistent.

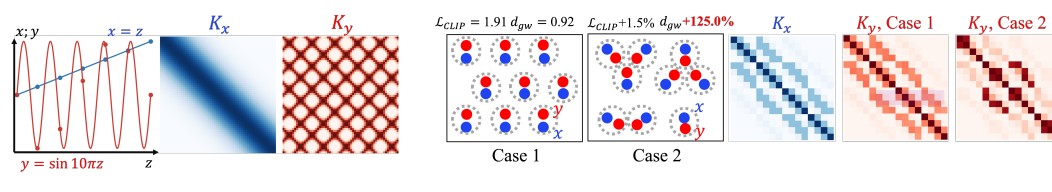

(a) 1D synthetic data           (b) 2D synthetic data

Figure 1: Structural alignment gap. (a) 1D synthetic data. Although $x$ and $y$ have high mutual information and thus a low InfoNCE loss $\mathcal{L}_c$, their intra-modal similarity matrices differ: (b) 2D synthetic data with instances (dashed gray circles). Blue and red denote two modalities. Pointwise correspondences are close in both cases (thus InfoNCE-like loss $\mathcal{L}_{\text{CLIP}}$ changes only $+1.5\%$), but the GW distance jumps by $+125\%$. Also, $K_y$ in Case 2 shows block structure absent from $K_x$.

We illustrate the gap with two synthetic examples in Fig. 1. **(a)** Let the latent variable be $z \sim \text{Uniform}[0, 1]$, from which we generate two modalities $x = z$ and $y = \sin(10\pi z)$. Although $x$ and $y$ are highly dependent, their intra-modal geometries differ markedly. In $x$, distances are simply $|x_i - x_j|$, whereas the high-frequency oscillation in $y$ disrupts local neighborhoods, so nearby $x$ can map to distant $y$, leading to dissimilar intra-modal similarity matrices. **(b)** modality $x$ forms a regular grid, while modality $y$ is either a globally shifted/noisy copy (*case 1*, left figure) or an unevenly shifted version that clusters points into triplets (*case 2*, right figure). Both cases preserve pointwise correspondences, leading to a lower InfoNCE loss. However, *case 2* distorts the global structure, which is reflected in block patterns in $K_y$ (kernel similarity matrix) that are absent in $K_x$.

This *structural alignment gap* is particularly critical in robotics, where multimodal sensor streams are neither i.i.d. nor structureless: trajectories form subclusters (Sermanet et al., 2017), contact events induce discontinuities (Stewart & Trinkle; Guo et al., 2023), and proprioceptive signals follow physical constraints (Lee et al., 2020; Welch & Bishop, 1995). Failing to account for these structures limits the effectiveness of learned representations for downstream robotic tasks.

To address the identified *structural alignment gap*, we introduce **UniOMA**—a **Uni**fied **O**ptimal-transport **M**ulti-modal structural **A**lignment framework that scales naturally to three or more modalities. UniOMA augments contrastive learning with a structure-aware regularization based on Gromov–Wasserstein (GW) distances and barycenters (Peyré et al., 2016; Gong et al., 2022). In our formulation, observations from each modality are represented as a metric space through intra-modal similarity matrices. A dynamic GW barycenter is then computed as the structural consensus across modalities, and each modality is softly aligned to this consensus by minimizing weighted GW distances. The modality weights are optimized end-to-end alongside encoder parameters, enabling adaptive contributions of different modalities to the structural consensus. This barycentric formulation avoids pairwise couplings across modalities, reducing the complexity from $O(M^2)$ to $O(M)$, where $M$ is the number of modalities, and thus scales naturally to three or more modalities.

In summary, our main contributions are:

- C1   We propose UniOMA, a structure-aware multimodal alignment framework based on Gromov–Wasserstein distance and barycenters, which naturally scales to 3+ modalities.
- C2   We identify and formalize the structural alignment gap, demonstrating why InfoNCE-style objectives fail to preserve intra-modal geometry, supported by synthetic analysis.

We evaluate UniOMA on diverse robotic benchmarks across vision, audio, tactile, force, and proprioception modalities, including robot state prediction, environment state prediction, and cross-modal retrieval. Comprehensive experiments show that UniOMA improves downstream performance and preserves intra-modal structural consistency across diverse modalities.

## 2   BACKGROUND AND RELATED WORK

In this section, we first introduce the background of contrastive learning-based multimodal alignment and review its extensions to settings with three or more modalities, highlighting their inherent

connections and limitations. We then briefly review existing approaches to multimodal representation learning in robotics, with a focus on multimodal fusion.

## 2.1 Alignment via InfoNCE and Extensions to More than Two Modalities

Unlike multimodal fusion (Lu et al., 2019; Li et al., 2019), which typically requires all modalities to be present at inference, alignment into a shared embedding space remains functional even if some modalities are missing, enabling zero-shot retrieval, generation, and modality completion (Jia et al., 2021). A representative example is CLIP (Radford et al., 2021), which trains modality-specific encoders $f_\theta^{(1)}, f_\theta^{(2)}$ using an InfoNCE-style objective to identify the correct cross-modal pair among $N$ candidates:

$$\ell_{\text{CLIP}}^{(1\to2)}(\theta) = -\frac{1}{N}\sum_{i=1}^{N}\log\frac{\exp\left(\text{sim}(\mathbf{z}_i^{(1)}, \mathbf{z}_i^{(2)})/\tau\right)}{\sum_{j=1}^{N}\exp\left(\text{sim}(\mathbf{z}_i^{(1)}, \mathbf{z}_j^{(2)})/\tau\right)}, \tag{1}$$

where $\text{sim}(\cdot,\cdot)$ is the similarity between the embeddings $\mathbf{z}^{(m)} = f_\theta^{(m)}(\mathbf{x}^{(m)}), m = 1, 2$ and $\tau$ denotes a temperature parameter. The final CLIP objective symmetrizes Eq. (1) by taking the average:

$$\mathcal{L}_{\text{CLIP}}^{(1,2)}(\theta) = \frac{1}{2}(\ell_{\text{CLIP}}^{(1\to2)}(\theta) + \ell_{\text{CLIP}}^{(2\to1)}(\theta)), \tag{2}$$

where $\mathcal{L}_{\text{CLIP}}^{(2\to1)}$ is the reverse direction $2 \to 1$. In general, this InfoNCE-based objective captures the statistical correlation, providing lower-bound of the mutual information (MI; Kraskov et al. (2004); Poole et al. (2019)) between the anchor modality 1 $\mathcal{X}^{(1)}$ and modality 2 $\mathcal{X}^{(2)}$

$$I(\mathcal{X}^{(1)}; \mathcal{X}^{(2)}) \geq \log N - 2\mathcal{L}_{\text{CLIP}}^{(1,2)}(\theta). \tag{3}$$

Despite their success, InfoNCE-like objectives reduce continuous similarity structure among samples into a binary signal (positive vs. negative), leading to the learned embedding space containing modality-wise co-located yet structurally isolated instances, neglecting intra-modal geometry.

Real-world systems, particularly in robotics, often involve three or more modalities. Aligning these multimodal sources within a shared embedding space enables richer cross-modal interactions. Existing approaches typically extend CLIP to three modalities by summing all pairwise contrastive losses (Tian et al., 2020; Girdhar et al., 2023; Akbari et al., 2021; Chen et al., 2023; Alayrac et al., 2020; Chen et al., 2021; Liu et al., 2024; Huang et al., 2023; Mai et al., 2022; Moon et al., 2022; Shvetsova et al., 2022; Xue et al., 2022; Guzhov et al., 2022):

$$\mathcal{L}_{\text{CMC}}^{(1,2,3)}(\theta) = \mathcal{L}_{\text{CLIP}}^{(1,2)}(\theta) + \mathcal{L}_{\text{CLIP}}^{(1,3)}(\theta) + \mathcal{L}_{\text{CLIP}}^{(2,3)}(\theta). \tag{4}$$

Such pairwise extensions neglect higher-order dependencies among modalities. To address this issue, Symile (Saporta et al., 2024) formulates triple-wise contrastive objectives as:

$$\mathcal{L}_{\text{Symile}}^{(1,2,3)}(\theta) = \frac{1}{3}[\ell^{(1\to2,3)}(\theta) + \ell^{(2\to1,3)}(\theta) + \ell^{(3\to1,2)}(\theta)]. \tag{5}$$

Here, $\ell^{(1\to2,3)}$ is the InfoNCE-like loss for one positive triple and $N-1$ negative triples given by

$$\ell^{(1\to2,3)}(\theta) = -\frac{1}{N}\sum_{i=1}^{N}\log\frac{\exp(\langle\mathbf{z}_i^{(1)}, \mathbf{z}_i^{(2)}, \mathbf{z}_i^{(3)}\rangle/\tau)}{\sum_{j=1}^{N}\exp(\langle\mathbf{z}_i^{(1)}, \mathbf{z}_j^{(2)}, \mathbf{z}_j^{(3)}\rangle/\tau)}, \tag{6}$$

where each term $\ell^{(1\to2,3)}$ compares one positive triple against $N-1$ negatives, $\langle\cdot,\cdot,\cdot\rangle$ is the coordinate-wise sum of the element-wise product. More recently, GRAM (Cicchetti et al., 2024) replaces the dot product similarity with the Gramian volume spanned by embeddings from multiple modalities, providing a higher-order, groupwise compatibility score (rather than pairwise similarity)

$$\mathcal{L}_{\text{GRAM}}^{(1,...,M)}(\theta) = \frac{1}{2}(\ell_{\text{D2A}}^{(1\to2,...,M)}(\theta) + \ell_{\text{A2D}}^{(1\to2,...,M)}(\theta)) + \lambda\ell_{\text{DAM}}(\theta), \tag{7}$$

$$\ell_{\text{D2A}}^{(1\to2,...,M)}(\theta) = -\frac{1}{N}\sum_{i=1}^{N}\log\frac{\exp(-\text{Vol}(\mathbf{z}_i^{(1)}, \mathbf{z}_i^{(2)}, \ldots, \mathbf{z}_i^{(M)})/\tau)}{\sum_{j=1}^{N}\exp(-\text{Vol}(\mathbf{z}_j^{(1)}, \mathbf{z}_i^{(2)}, \ldots, \mathbf{z}_i^{(M)})/\tau)}, \tag{8}$$

$$\ell_{\text{A2D}}^{(1\to2,...,M)}(\theta) = -\frac{1}{N}\sum_{i=1}^{N}\log\frac{\exp(-\text{Vol}(\mathbf{z}_i^{(1)}, \mathbf{z}_i^{(2)}, \ldots, \mathbf{z}_i^{(M)})/\tau)}{\sum_{j=1}^{N}\exp(-\text{Vol}(\mathbf{z}_i^{(1)}, \mathbf{z}_j^{(2)}, \ldots, \mathbf{z}_j^{(M)})/\tau)}. \tag{9}$$

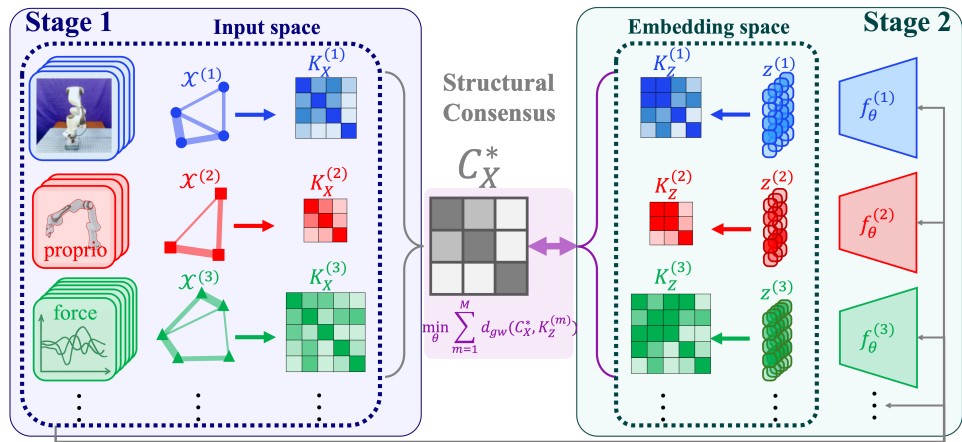

Figure 2: **UniOMA in two stages.** *Stage 1* (left): for each modality $\mathcal{X}^{(m)}$ we form an input-space similarity matrix $\mathbf{K}_{\mathbf{x}}^{(m)}$ and estimate a GW barycenter $\mathbf{C}_{\mathbf{x}}^*$ as the *structural consensus*. *Stage 2* (right): encoders produce embeddings $\mathbf{z}^{(m)}$ inducing $\mathbf{K}_{\mathbf{z}}^{(m)}$, which are aligned to the consensus by minimizing $\sum_m \lambda_m d_{gw}(\mathbf{C}_{\mathbf{x}}^*, \mathbf{K}_{\mathbf{z}}^{(m)})$ (with a standard contrastive loss; omitted). Aligning each modality to a single consensus avoids pairwise $O(M^2)$ couplings and scales to $M \geq 3$.

where $\mathcal{L}_{\text{D2A}}, \mathcal{L}_{\text{A2D}}$ are the GRAM contrastive loss (data-to-anchor for D2A and anchor-to-data for A2D) with modality 1 as the anchor. $\mathcal{L}_{\text{DAM}}$ is the data-caption matching loss to match the modality labels (Cicchetti et al., 2024). $\text{Vol}(\cdot, \ldots, \cdot)$ is the volume of the $M$-dimensional parallelotope formed by the embedding vectors $\mathbf{z}^{(m)}$.

These methods mark progress toward multi-modal ($M > 2$) alignment but still remain limited to instance-level dependencies, overlooking intra-modal structure. Zhu & Luo (2024) address this by adding an optimal transport (OT; Villani et al. (2008)) regularizer to enforce cross-modal consistency. Yet, their approach still treats modalities as holistic distributions, ignoring relational structures within each modality, and applies OT directly on embeddings rather than raw data geometry, limiting interpretability and flexibility.

### 2.2 MULTIMODAL REPRESENTATION LEARNING IN ROBOTICS

Robotics is inherently multimodal: vision, force–torque, tactile sensing, and proprioception provide complementary views of the robot–environment system. While multimodal representation learning has been extensively studied in vision–language settings, its exploration in robotics remains limited. Existing work, including the recent Vision–Language–Action (VLA) model, has primarily focused on modality fusion or transfer (Lee et al., 2019a;b; Shridhar et al., 2020; Brohan et al., 2022; Driess et al., 2023; Kim et al., 2024; Octo Model Team et al., 2024; Intelligence et al., 2025).

By comparison, alignment of robotic perception modalities into a shared space remains underexplored. Recent efforts (Wojcik et al., 2024; Dutta et al., 2024) demonstrate cross-modal retrieval and perception, while Zambelli et al. (2021); Sermanet et al. (2017) demonstrate how cross-modal or cross-temporal alignment can yield transferable representations. These developments underscore that robot perception data is highly structured (trajectories, contact events, physical constraints), motivating alignment methods that preserve intra-modal geometry across modalities rather than relying solely on fusion.

## 3 METHOD

Our proposed UniOMA aligns three or more heterogeneous modalities by preserving both statistical correspondence and structural coherence across modalities. Leveraging the optimal transport geometry, UniOMA augments contrastive-based binary instance-wise correlations (positive or neg-

ative pair) with structural properties by minimizing the Gromov-Wasserstein (GW) distance across the modalities. In this section, we first explain the multimodal alignment problem, and then we define the intra-modal structure information and the cross-modal structure consensus, followed by the definition of the UniOMA objective and the alignment algorithm.

### 3.1 PROBLEM STATEMENT

Let $\mathcal{X}^{(1)}, \mathcal{X}^{(2)}, \ldots, \mathcal{X}^{(M)}$ denote $M$ modalities. The goal of multimodal alignment is to learn modality-specific encoders $f^{(m)} : \mathcal{X}^{(m)} \to \mathbb{R}^d$, $m = 1, \ldots, M$ that project inputs $\mathbf{x}^{(m)} \in \mathcal{X}^{(m)}$ into a shared latent space $\mathbf{z}^{(m)} = f^{(m)}(\mathbf{x}^{(m)}) \in \mathbb{R}^d$. The key objective is that embeddings of the same underlying instance across modalities map to nearby latent vectors, i.e., $\mathbf{z}_i^{(1)} \approx \mathbf{z}_i^{(2)} \approx \cdots \approx \mathbf{z}_i^{(M)}$. In robotics, the modalities $\mathcal{X}^{(m)}$ may include vision (third-person or wrist-mounted), audio commands, force–torque signals, proprioception (joints, inertial measurement unit (IMU), end-effector pose), tactile sensing, and environment states (e.g., object pose). Aligning them into a shared latent space enables cross-modal reasoning and zero-shot transfer: for example, vision of an end-effector trajectory should yield embeddings consistent with the same trajectory from proprioception or touch. Such unified representations enable downstream tasks such as robot/environment state prediction, action prediction or generation, and modality completion when certain sensor streams are missing.

### 3.2 GROMOV-WASSERSTEIN DISTANCE

The Gromov–Wasserstein (GW) distance (Peyré et al., 2016; Gong et al., 2022) is a natural extension of Optimal Transport (OT) (Villani et al., 2008) to settings where distributions lie in different metric spaces. The classic OT problem seeks the minimum cost of transporting one probability measure into another within the same metric space. Given two measures $\mu$ and $\nu$ and a cost function $c : \mathcal{X} \times \mathcal{X} \to \mathbb{R}$, the Kantorovich formulation of the OT problem is

$$d_w(\mu, \nu) := \inf_{\pi \in \Pi(\mu, \nu)} \int_{\mathcal{X} \times \mathcal{X}} c(\mathbf{x}, \mathbf{y}) d\pi(\mathbf{x}, \mathbf{y}), \tag{10}$$

where $\pi$ is a transport plan with marginals $\mu$ and $\nu$. When both measures are supported on the same space $\mathcal{X}$, $c(\cdot, \cdot)$ is a distance metric (e.g., $\ell_2$), and Eq. 10 defines the Wasserstein distance.

However, in multimodal learning the two distributions often live in different spaces (e.g., images vs. tactile signals). In such cases, defining a cross-modal cost $c(\mathbf{x}, \mathbf{y})$ is generally impossible. The GW distance addresses this by replacing the direct cross-modal cost with a relational cost that compares intra-modal similarities.

**Definition 1** (Gromov-Wasserstein Distance). *Let $\mathcal{X}_{d_\mathbf{x}, \mu}$ and $\mathcal{Y}_{d_\mathbf{y}, \nu}$ be two metric–measure spaces (mm-spaces), with distance metrics $d_\mathbf{x}$, $d_\mathbf{y}$ and probability measures $\mu, \nu$. The GW distance between them is defined as:*

$$d_{gw}(\mu, \nu) := \inf_{\pi \in \Pi(\mu, \nu)} \int_{\mathcal{X}^2 \times \mathcal{Y}^2} c\left(d_\mathbf{x}(\mathbf{x}, \mathbf{x}'), d_\mathbf{y}(\mathbf{y}, \mathbf{y}')\right) d\pi(\mathbf{x}, \mathbf{y}) d\pi(\mathbf{x}', \mathbf{y}'),$$

*where $c\left(d_\mathbf{x}(\mathbf{x}, \mathbf{x}'), d_\mathbf{y}(\mathbf{y}, \mathbf{y}')\right)$ is relational distance measuring the discrepancy between the sample pairs $(\mathbf{x}, \mathbf{x}')$ and $(\mathbf{y}, \mathbf{y}')$.*

Intuitively, minimizing GW distance aligns two distributions by matching their relational geometry (pairwise structures), rather than raw coordinates. This is crucial in robotics, where modalities such as vision and force–torque are in incomparable metric spaces but have meaningful internal geometries. For discrete samples, consider the two mm-spaces $\mathcal{X} = \{\mathbf{x}_i\}_{i=1}^I$ and $\mathcal{Y} = \{\mathbf{y}_j\}_{j=1}^J$ with uniform sample distributions $\hat{\mathbf{p}}_\mathbf{x} = \frac{1}{I} \mathbf{1}_I$ and $\hat{\mathbf{p}}_\mathbf{y} = \frac{1}{J} \mathbf{1}_J$, we calculate the empirical GW distance (Gong et al., 2022) in the following definition.

**Theorem 1** (Empirical GW Distance). *Let the kernel matrices $\mathbf{K}_\mathbf{x} \in \mathbb{R}^{I \times I}$ and $\mathbf{K}_\mathbf{y} \in \mathbb{R}^{J \times J}$ be the similarity matrices conducted by the samples $\mathbf{x}, \mathbf{y}$ from two mm-spaces $\mathcal{X}, \mathcal{Y}$, the empirical GW distance between the samples is:*

$$\hat{d}_{gw}(\mathbf{K}_\mathbf{x}, \mathbf{K}_\mathbf{y}) := \max_{\mathbf{T} \in \Pi(\hat{\mathbf{p}}_\mathbf{x}, \hat{\mathbf{p}}_\mathbf{y})} \mathrm{tr}(\mathbf{K}_\mathbf{x}^\top \mathbf{T}^\top \mathbf{K}_\mathbf{y} \mathbf{T}),$$

*where $\mathbf{T}$ is the doubly-stochastic matrix to model the transport between the two sets of samples.*

See Appx. A.5 for the proof. In practice, we estimate $\mathbf{T}^*$ via iterative OT solvers (Alg. 2), and compute $\hat{d}_{gw}(\mathbf{K_x}, \mathbf{K_y}) = \mathrm{tr}(\mathbf{K_x}^\top \mathbf{T}^{*\top} \mathbf{K_y} \mathbf{T}^*)$. This formulation enables cross-modal alignment directly from intra-modal similarity structures, without the need of an explicit cross-modal cost function or extra neural potential models (Korotin et al., 2022b;a; Arjovsky et al., 2017).

### 3.3 Structural Consensus

To preserve intra-modal structure during alignment, we treat each modality $\mathcal{X}^{(m)}$ as a metric space and represent its geometry via a kernel matrix $\mathbf{K_x}^{(m)} \in \mathbb{R}^{N_m \times N_m}$, where $(K_{\mathbf{x}}^{(m)})_{ij} = \mathrm{sim}(\mathbf{x}_i^{(m)}, \mathbf{x}_j^{(m)})$ encodes the pairwise similarity between samples. Such kernel matrices provide a unified representation of relational structure across heterogeneous modalities, independent of raw dimensionality. The construction of $\mathbf{K_x}^{(m)}$ depends on the modality: for visual signals (e.g., RGB or depth), we embed inputs with a pretrained encoder and compute similarities using an RBF kernel; for sequential or time-series modalities common in robotics (e.g., force–torque), we adopt a time-series clustering kernel (TCK; Mikalsen et al. (2018)) to better capture temporal structure. Additional details are provided in Appx. B.3.

The central idea is to identify a structural consensus: a latent geometry that captures the common similarity patterns across all modalities. Formally, we define it as a Gromov–Wasserstein (GW) barycenter (Gong et al., 2022) of the intra-modal structures.

**Definition 2** (Structural Consensus of Multimodal Data). *Given intra-modal kernel matrices* $\{\mathbf{K_x}^{(m)}\}_{m=1}^M$, *the structural consensus is defined as the barycenter:*

$$\mathbf{C_x}^* = \arg \min_{\mathbf{C_x} \in \mathcal{M}} \sum_{m=1}^{M} \lambda_m \cdot d_{gw}(\mathbf{C_x}, \mathbf{K_x}^{(m)}), \tag{11}$$

*where $\mathcal{M}$ denotes the space of symmetric positive definite (SPD) matrices, $d_{gw}$ is the GW distance (Def. 1), and $\lambda_m$ are learnable modality weights.*

Practically, $\mathbf{C_x}^*$ is estimated via an iterative optimization scheme (Alg. 3 in Appx. B.2). During training, we align each modality by minimizing the GW discrepancy between its embedding-induced kernel $\mathbf{K_z}^{(m)}$ and the consensus $\mathbf{C_x}^*$, as described in the next section.

### 3.4 UniOMA Objective and Algorithm

Given the batch-wise structural consensus $\mathbf{C_x}^*$ in Sec. 3.3, UniOMA augments a standard contrastive term with a structure-aware regularizer

$$\mathcal{L}_{\mathrm{UniOMA}}(\theta) = \mathcal{L}_{\mathrm{c}}(\theta) + \alpha \sum_{m=1}^{M} \lambda_m \cdot d_{gw}(\mathbf{C_x}^*, \mathbf{K_z}^{(m)}), \tag{12}$$

where $\mathbf{K_z}^{(m)}$ is the embedding-space similarity matrix of $\mathbf{z}^{(m)} = f_\theta^{(m)}(\mathbf{x}^{(m)})$. The scalar $\alpha$ balances contrastive discrimination and structural coherence, and the learnable weights $\{\lambda_m\}$ quantify each modality's contribution to the consensus. Implementation details for estimating $\mathbf{C}_x^*$ and evaluating $d_{gw}(\cdot, \cdot)$ are in Appx. B.2–A.5 (see also Fig. 2).

**Why this design?** (1) **Scalable** to $M \geq 3$. Aligning every modality to one consensus avoids $O(M^2)$ pairwise couplings. (2) **Flexible** to heterogeneous and asynchronous modalities. GW distance compares intra-modal similarity matrices, not raw coordinates, thus is naturally robust to modalities with different dimensionalities. Also, GW barycenter naturally handles unequal sample counts across modalities, which is particularly advantageous in robot perception. We empirically validate (3) in Sec. 4.6.

---

**Algorithm 1** UniOMA Training($\{\mathcal{X}^{(m)}\}_{m=1}^M, \gamma, \alpha$)

---

**Input:** Multimodal dataset $\{\mathcal{X}^{(m)}\}_{m=1}^M$, learning rate $\gamma$, structural weight $\alpha$, entropy weight $\alpha'$

Initialize encoders $\{f^{(m)}(\cdot)\}_{m=1}^M$, modality weights $\{\lambda_m\}_{m=1}^M$,

**while** *not converged* **do**

> // Stage 1: structural consensus estimation
>
> Sample a batch $\{\mathbf{x}_i^{(m)}\}_{i=1}^{N_m}$ for each modality $\{\mathcal{X}^{(m)}\}_{m=1}^M$
> **for** $m \leftarrow 1$ **to** $M$ **do**
> > Compute the structural information $\mathbf{K}_{\mathbf{x}}^{(m)} \in \mathbb{R}^{N_m \times N_m}$ for the batch $\{\mathbf{x}_i^{(m)}\}_{i=1}^{N_m}$
>
> Estimate the structural consensus $\mathbf{C}_{\mathbf{x}}^*$ via Alg. 3

> // Stage 2: alignment update
>
> $\mathbf{z}_i^{(m)} \leftarrow f_\theta^{(m)}(\mathbf{x}_i^{(m)})$ for all $i, m$
> **for** $m \leftarrow 1$ **to** $M$ **do**
> > $\mathbf{T}^{(m)^*} \leftarrow \texttt{OTEstimation}(\mathbf{C}_{\mathbf{x}}^*, \mathbf{K}_{\mathbf{z}}^{(m)})$ via Alg. 2
> > $\hat{d}_{gw}(\mathbf{C}_{\mathbf{x}}^*, \mathbf{K}_{\mathbf{z}}^{(m)}) \leftarrow \text{tr}((\mathbf{C}_{\mathbf{x}}^*)^\top (\mathbf{T}^{(m)^*})^\top \mathbf{K}_{\mathbf{z}}^{(m)} \mathbf{T}^{(m)^*})$
>
> Compute the contrastive learning loss $\mathcal{L}_c$
> $\mathcal{L}_{\text{UniOMA}}(\theta) \leftarrow \mathcal{L}_c(\theta) + \alpha \sum_{m=1}^M \lambda_m \hat{d}_{gw}(\mathbf{C}_{\mathbf{x}}^*, \mathbf{K}_{\mathbf{z}}^{(m)})$
> $\theta \leftarrow \theta - \alpha \nabla_\theta \mathcal{L}_{\text{UniOMA}}$
> $\lambda_m \leftarrow \lambda_m - \alpha \nabla_{\lambda_m} \mathcal{L}_{\text{UniOMA}}$ for $m = 1, \dots, M$

**return** $\{f_\theta^{(m)}\}_{m=1}^M, \{\lambda_m\}_{m=1}^M$

---

The training procedure is summarized in Alg. 1. Each iteration proceeds in two stages:

> **Stage 1 (Consensus Estimation):** Compute kernel matrices $\mathbf{K}_{\mathbf{x}}^{(m)}$ from a mini-batch using modality-specific similarity measures (e.g., RBF kernel for images, TCK for time series), then estimate the batch-wise consensus $\mathbf{C}_{\mathbf{x}}^*$ via an iterative GW barycenter algorithm (Appx. B.2).

> **Stage 2 (Alignment Update):** Encode the same mini-batch into $\mathbf{z}^{(m)}$, form kernel matrices $\mathbf{K}_{\mathbf{z}}^{(m)}$, and compute their GW distances to the consensus. The UniOMA loss is then minimized by stochastic gradient descent, jointly updating encoder parameters $\theta$ and modality weights $\lambda_m$.

# 4 EXPERIMENTS

We evaluate UniOMA on four multimodal robot perception settings: (i) VFD (Vision–Force–Depth) from the VisionTouch dataset (Lee et al., 2019b; Liang et al., 2021); (ii) VFP (Vision–Force–Proprioception) from the same source; (iii) MuJoCo Push (Lee et al., 2020; Todorov et al., 2012) (Vision–Force–End-effector pose); and (iv) VAT (Vision–Audio–Tactile) derived from ObjectFolder 2.0 (Gao et al., 2022; Wojcik et al., 2024). Downstream tasks include regression, classification, and cross-modal retrieval.

## 4.1 TASKS AND DATASETS

**VFD (Vision–Force–Depth).** We evaluate two tasks: (1). Next-step end-effector orientation prediction (regression, 4D): Inputs are third-person RGB ($[b \times 3 \times 128 \times 128]$), force–torque histories ($[b \times 32 \times 6]$), and depth ($[b \times 1 \times 128 \times 128]$). (2). Trajectory-pair discrimination (classification, binary): given a pair of triplets (vision-force-depth), identify whether the pair is from the the same trajectory. We report Top-1 accuracy in Table 1.

**VFP (Vision–Force–Proprioception).** We evaluate next-step contact prediction (classification, binary). Inputs are RGB, force–torque histories, and end-effector pose ($[b \times 7]$). We classify whether the end-effector is in contact to the object.

Table 1: Comparative results on downstream tasks (regression, classification, and cross-modal retrieval). Performance is measured by MSE ($\times 10^{-3}$ $\downarrow$), Top-1 Acc. (% $\uparrow$), and MAP ($\uparrow$). Arrows denote retrieval direction. Gray rows correspond to baselines augmented with our GW regularizer. Overall, our method consistently improves its corresponding baselines across most tasks, and all methods achieving the best performance for each task are UniOMA variants (highlighted in brown).

| Method | Regression $\downarrow(\times 10^{-3})$ | | Classification $\uparrow(\%)$ | | VAT MAP Score $\uparrow$ | | |
| | V&F&D | MuJoCo | V&F&D | V&F&P | Vis→Aud | Vis→Tact | Tact→Aud |
|---|---|---|---|---|---|---|---|
| Pairwise | $1.27_{\pm 0.14}$ | $0.44_{\pm 0.07}$ | $89.59_{\pm 0.05}$ | $94.51_{\pm 0.02}$ | $0.25_{\pm 0.07}$ | $0.41_{\pm 0.11}$ | $0.10_{\pm 0.01}$ |
| Pairwise+OT | $1.26_{\pm 0.11}$ | $0.40_{\pm 0.07}$ | $92.41_{\pm 0.02}$ | $94.66_{\pm 0.02}$ | $\mathbf{0.37}_{\pm 0.05}$ | $0.58_{\pm 0.04}$ | $0.09_{\pm 0.01}$ |
| Pairwise+GW | $1.22_{\pm 0.12}$ | $\mathbf{0.38}_{\pm 0.09}$ | $92.44_{\pm 0.02}$ | $94.68_{\pm 0.03}$ | $0.36_{\pm 0.05}$ | $0.60_{\pm 0.03}$ | $\mathbf{0.12}_{\pm 0.02}$ |
| Symile | $2.81_{\pm 0.10}$ | $0.28_{\pm 0.04}$ | $90.02_{\pm 0.04}$ | $\mathbf{93.94}_{\pm 0.06}$ | $0.10_{\pm 0.02}$ | $\mathbf{0.21}_{\pm 0.05}$ | $0.08_{\pm 0.01}$ |
| Symile+GW | $\mathbf{2.15}_{\pm 0.08}$ | $\mathbf{0.23}_{\pm 0.02}$ | $\mathbf{92.81}_{\pm 0.02}$ | $93.87_{\pm 0.03}$ | $\mathbf{0.13}_{\pm 0.03}$ | $0.15_{\pm 0.03}$ | $\mathbf{0.14}_{\pm 0.03}$ |
| GRAM | $3.37_{\pm 0.09}$ | $0.52_{\pm 0.07}$ | $92.47_{\pm 0.04}$ | $93.65_{\pm 0.05}$ | $0.13_{\pm 0.02}$ | $0.34_{\pm 0.05}$ | $0.15_{\pm 0.01}$ |
| GRAM+GW | $\mathbf{2.31}_{\pm 0.05}$ | $\mathbf{0.30}_{\pm 0.06}$ | $\mathbf{93.30}_{\pm 0.01}$ | $\mathbf{93.91}_{\pm 0.04}$ | $\mathbf{0.79}_{\pm 0.10}$ | $\mathbf{0.58}_{\pm 0.04}$ | $\mathbf{0.16}_{\pm 0.01}$ |
| CoMM | $1.51_{\pm 0.05}$ | $0.26_{\pm 0.04}$ | $92.39_{\pm 0.01}$ | $94.13_{\pm 0.03}$ | — | — | — |

Table 2: **Scalability analysis with 4–7 modalities.** We report trajectory-pair classification accuracy (mean $\pm$ std over 10 seeds) together with wall-clock time per epoch for all three methods: Pairwise contrastive learning, Pairwise+OT, and Pairwise+GW (UniOMA). UniOMA achieves consistently higher accuracy and becomes faster than OT when $M \geq 6$.

| Modality Combination | $M$ | Pairwise | | Pairwise+OT | | Pairwise+GW | |
| | | Acc. | Time | Acc. | Time | Acc. | Time |
|---|---|---|---|---|---|---|---|
| V+F+P+D | 4 | $89.94_{\pm 0.03}$ | $110.38_{\pm 1.74s}$ | $92.07_{\pm 0.03}$ | $135.57_{\pm 2.92s}$ | $\mathbf{92.39}_{\pm 0.02}$ | $201.36_{\pm 7.61s}$ |
| V+F+P+D+A | 5 | $90.72_{\pm 0.03}$ | $129.44_{\pm 1.92s}$ | $92.51_{\pm 0.03}$ | $178.63_{\pm 3.11s}$ | $\mathbf{93.04}_{\pm 0.02}$ | $225.89_{\pm 5.44s}$ |
| V+F+P+D+A+C | 6 | $89.12_{\pm 0.04}$ | $150.77_{\pm 2.51s}$ | $91.03_{\pm 0.03}$ | $268.41_{\pm 6.83s}$ | $\mathbf{92.11}_{\pm 0.03}$ | $248.52_{\pm 6.12s}$ |
| V+F+P+D+A+C+O | 7 | $87.95_{\pm 0.05}$ | $171.42_{\pm 3.12s}$ | $89.84_{\pm 0.04}$ | $382.77_{\pm 10.44s}$ | $\mathbf{91.02}_{\pm 0.03}$ | $273.36_{\pm 7.40s}$ |

**MuJoCo Push.** A planar pushing task with a Franka Emika Panda arm interacting with a puck. Inputs are low-resolution gray-scale image ($[b \times 1 \times 32 \times 32]$), current force–torque ($[b \times 6]$), and end-effector pose ($[b \times 7]$). The task is to predict the next-step object's 2-D position on the table.

**VAT (Vision–Audio–Tactile).** We evaluate **cross-modal retrieval** using mean average precision (MAP). Queries and retrievals are built across {Vis, Aud, Tact}; we report direction-specific MAP (e.g., Vis→Tact). The dataset provides per-object visual, sound, and tactile observations.

**Scalability to 4–7 Modalities.** To evaluate the scalability of UniOMA beyond three modalities, we introduce a new downstream classification task on the Vision&Touch dataset training on 4, 5, 6, and 7 modalities (vision, force, proprioception, depth, action, contact, and optical flow). The task is to classify whether two multimodal/single-modal samples originate from the same trajectory.

### 4.2 IMPLEMENTATION DETAILS

Encoders, optimizer, temperature, and schedules are shared across methods (fusion heads differ in CoMM). We compute input-space kernels $\{\mathbf{K}_\mathbf{x}^{(m)}\}$ (RBF for images with tuned $\gamma$; TCK for time-series/force; RBF for other signals) and estimate the batch-wise consensus $\mathbf{C}_\mathbf{x}^*$ using iterative barycenter updates (Appx. B.2). We then align embedding-space kernels $\{\mathbf{K}_\mathbf{z}^{(m)}\}$ to $\mathbf{C}_\mathbf{x}^*$ via the UniOMA loss. Hyperparameters, TCK settings, and convergence diagnostics are detailed in Appx. B.3–B.3.

### 4.3 RESULTS: COMPARISONS ON DOWNSTREAM TASKS

We compare against: (i) **Pairwise** (CMC) (Tian et al., 2020) using summed pairwise InfoNCE; (ii) **Symile** (Saporta et al., 2024) using triple-wise InfoNCE variants; (iii) **GRAM** (Cicchetti et al., 2024)

using Gramian volume similarity for $M \geq 3$; and (iv) **CoMM** (Dufumier et al., 2024) as a strong fusion-based baseline. For (i)–(iii) we also report " +GW" variants by adding our GW regularizer to show the marginal value of structural alignment. We match optimizer, batch size, temperature, and training epochs across comparable methods; see Appx. B.3.

Table 1 summarizes results across the 3-modality tasks in Sec. 4.1. Overall, UniOMA with its GW-augmented variants consistently outperform purely contrastive objectives. In particular, adding our GW regularizer ( +GW) yields stable gains across all objectives, confirming that structure-aware alignment provides benefits orthogonal to instance discrimination. In the two cells where a baseline is slightly higher (Symile on VFP classification and Vis→Tact), the GW term trades a bit of contrastive correlation for structural coherence. All hyperparameters were kept fixed across methods.

## 4.4 RESULTS: EFFICIENCY AND SCALABILITY

Table 2 reports the results of the new-introduced task with 4-7 modalities. Because additional modalities introduce greater distributional heterogeneity, aligning them becomes increasingly challenging. As a result, traditional pairwise contrastive and OT-based approaches do not exhibit improved classification accuracy as the number of modality increases. In contrast, our GW-based method maintains stable performance and consistently achieves the highest accuracy compare to the pairwise / pairwise+OT baseline, demonstrating better scalability in high-modality scenarios. A detailed description of the 4–7 modality setup is provided in Appx. C.

UniOMA is designed to avoid the quadratic complexity inherent in pairwise multimodal alignment, which computes $O(M^2)$ cross-modal couplings across $M$ modalities. UniOMA aligns each modality independently to a learned structural consensus, yielding linear complexity $O(M)$. We measure runtime and peak memory as a function of the number of modalities (3–7). Table 2 shows wall-clock time per epoch of UniOMA grows approximately linearly with the number of modalities $M$, while pairwise and OT-based baselines have quadratic scaling. For $M \geq 6$, UniOMA becomes strictly faster per epoch than the pairwise+OT baseline, while peak memory usage remains identical. We also observe in Fig. 6 that the GW barycenter converges stably with $T_{\max} = 5$ iterations across all settings (with runtime mildly increased as shown in Table 2). Ablations with $T_{\max} \in \{2, 5, 10\}$ in Table 3 confirm that performances are stable with respect to solver iterations. These results indicate that minibatch GW inference introduces only moderate overhead and does not impair training practicality.

## 4.5 RESULTS: MODALITY WEIGHTS

UniOMA learns modality weights $\{\lambda_m\}$ that quantify each modality's contribution to the consensus (Appx. B.2). Fig. 3 shows that vision dominates VAT retrieval (high discriminative content); proprioception dominates VFP contact prediction (contact reasoning); depth is critical for VFD orientation regression, and force contributes marginally.

## 4.6 ABLATION STUDY: UNEQUAL MODALITY SAMPLING

To evaluate UniOMA's robustness to realistic asynchrony in robot perception, we perform an ablation on the VFD classification task. Specifically, we downsample one modality per batch (vision, force, or depth) by a factor of two, e.g. we downsample one modality (e.g., $b=32$) while keeping others at $b=64$, inducing unequal sample counts and breaking strict one-to-one pairing across modalities. We compare UniOMA against its contrastive-only variant (pairwise vs. pairwise+GW). Fig. 3(f) shows that UniOMA (Pairwise+GW) outperforms the contrastive-only baseline (Pairwise) across all downsampling cases. This confirms that aligning each modality to the GW barycenter consensus, rather than enforcing pairwise matches, enables the model to effectively leverage heterogeneous modality even under sampling-rate mismatch.

**Interpretability.** Beyond accuracy, UniOMA provides insights into modality importance through its learned weights. Figure 3(e) visualizes the weight distributions under each downsampling setting, showing how the framework adaptively shifts reliance toward intact modalities while still retaining useful signal from the under-sampled one. For comparison, Figure 3(a-d) aggregates the learned weights across the four benchmark datasets (VFP, VFD, MuJoCo, VAT), illustrating task-dependent

modality dominance. These results highlight UniOMA's ability to not only maintain structural alignment under unequal sampling but also to yield interpretable modality relevance.

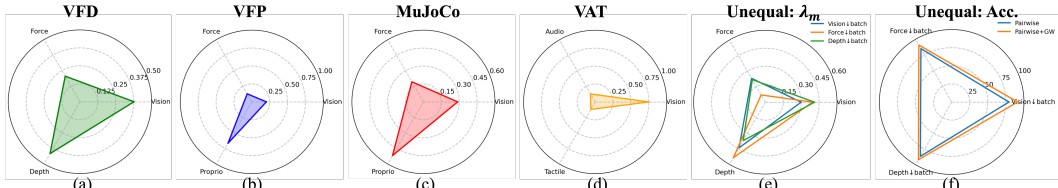

Figure 3: (a–d) Final learned modality weights $\{\lambda_m\}$ for each task (VFD, VFP, MuJoCo Push, VAT). Each radar chart shows per-modality weights that sum to 1, highlighting dataset-specific salience (e.g., depth in VFD, proprioception in VFP) and the interpretability of UniOMA's structural-consensus weighting. (e) ablation on VFD. One modality is downsampled by $\times\frac{1}{2}$ per batch. The plot shows UniOMA's adaptive redistribution of $\{\lambda_m\}$ toward intact modalities while retaining signal from the undersampled one. (f) Accuracy under the same ablation (Top-1, %). Pairwise vs. Pairwise+GW (UniOMA). The outer polygon indicates consistent gains from the GW regularizer across all downsampled cases.

## 5 DISCUSSION AND CONCLUSION

**Interpreting the GW barycenter and its applicability.** Our visualizations (Appx. G) show that batch-wise GW barycenters recover intuitive geometric patterns across modalities, reflecting that GW aligns *structural relations* rather than raw features. This behavior is well suited to robotics, where trajectories naturally form meaningful intra-modal graphs. In structurally poor domains such as vision–language–audio with i.i.d. samples, however, useful barycenters require constructing richer intra-modal graphs first—an explicit limitation and a promising extension for more general multimodal learning.

**Shared vs. modality-specific information under alignment.** Our theory (Appx. A) and experiments support a classical view from multimodal information bottleneck and Partial Information Decomposition: alignment should extract only the *shared* structure while preserving modality-specific (high-frequency) information. UniOMA achieves this by constraining embeddings only through low-frequency consensus, leaving modality-specific components to be shaped by the contrastive objective. This also clarifies a limitation in vision–language settings: most VLM datasets are instance-wise and lack trajectory-style intra-modal geometry, making GW barycenters less meaningful without an additional graph-construction step.

**Conclusion.** We revisit multimodal alignment through the lens of *structural* consistency: while pointwise correspondences are statistically strong in existing alignment methods, the intra-modal geometries can disagree across modalities. UniOMA closes this gap by combining standard contrastive learning with a GW-barycenter regularizer that aligns 3+ modalities to a shared structural consensus. Across VFP, VFD, MuJoCo Push, VAT, and 4–7 modality settings, UniOMA improves regression, classification, and cross-modal retrieval while learning interpretable, dataset-specific modality weights. Limitations include the additional computational cost of barycentric GW updates and sensitivity to kernel choices. Our mini-batch barycenter and kernel ablations mitigate these costs but do not fully remove them. Promising future directions include large-scale real-robot alignment under heterogeneous sampling rates and extensions to asymmetric similarity kernels (e.g., directed or causal structures).

## 6 REPRODUCIBILITY STATEMENT

For the method's implementation, we include the details in B.3. For the used datasets, Appx. C provides a complete description of preprocessing and splits for VFP, VFD, MuJoCo Push, VAT, and the 4-7 modality task. For theory, Appx. A.5–B.2 contain clear assumptions, derivations, and algorithmic details used in UniOMA.

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

# A   ADDITIONAL THEORETICAL DISCUSSION

In this appendix, we give a theoretical view of our structural alignment, which explains its feasibility: (i) the GW-based term acts locally like a graph-smoothing force on the shared, low-frequency structure of the embeddings; and (ii) this does *not* force a information-rich modality (e.g., image) to discard its modality-specific, high-frequency information. To specific, we discuss in three steps: firstly, near the structural consensus $\mathbf{C}_{\mathbf{x}}^*$, the gradient of the GW distance is aligned with the gradient of the Dirichlet energy (Belkin & Niyogi, 2003; Chung, 1996) on the Laplacian (Lemma 1); we then analyze that the Dirichlet energy is spectrally biased towards low-frequency eigenmodes (Lemma 2); finally, we combine these findings to argue that our UniOMA regularizer aligns only the shared low-frequency geometry while leaving higher-frequency components available to encode modality-specific information (Theorem 2).

## A.1   SETUP

Let $n$ be the batch size and $\mathbf{Z} \in \mathbb{R}^{n \times d}$ the embedding matrix of one modality, with rows $\mathbf{z}_1, \ldots, \mathbf{z}_n$. We construct a similarity matrix

$$\mathbf{K}_{\mathbf{z}}(i, j) = k(\|\mathbf{z}_i - \mathbf{z}_j\|^2),$$

where $k : \mathbb{R}_+ \to \mathbb{R}_+$ is a kernel that is strictly decreasing (e.g., an RBF kernel). Let $\mathbf{C}_{\mathbf{x}}^* \in \mathbb{R}^{n \times n}$ be a fixed barycenter similarity matrix (our structural consensus), and let $\mathbf{L}^*$ be the associated normalized Laplacian:

$$\mathbf{L}^* = \mathbf{D} - \mathbf{C}_{\mathbf{x}}^*,$$

where $\mathbf{D} = \mathbf{I}$ is the degree matrix with $\mathbf{D}_{ii} = \sum_j \mathbf{C}_{\mathbf{x}}^*(i, j)$. The Dirichlet energy of $\mathbf{Z}$ on this consensus geometry is

$$E_{\mathrm{Dir}}(\mathbf{Z}) = \mathrm{tr}(\mathbf{Z}^\top \mathbf{L}^* \mathbf{Z}) = \frac{1}{2} \sum_{i,j} \mathbf{L}^*(i, j) \|\mathbf{z}_i - \mathbf{z}_j\|^2,$$

For the structural term, we consider the squared-loss Gromov–Wasserstein discrepancy between $\mathbf{K}_{\mathbf{z}}$ and $\mathbf{C}_{\mathbf{x}}^*$:

$$d_{\mathrm{GW}}^2(\mathbf{K}_{\mathbf{z}}, \mathbf{C}_{\mathbf{x}}^*) = \min_{\pi \in \Pi(p,q)} \sum_{i,j,k,\ell} \left( \mathbf{K}_{\mathbf{z}}(i, j) - \mathbf{C}_{\mathbf{x}}^*(k, \ell) \right)^2 \pi_{ik} \pi_{j\ell}, \tag{13}$$

where $\Pi(p, q)$ is the set of transport plans with fixed marginals $p, q$. In the batch setting we consider here, $p = q = \frac{1}{n}\mathbf{1}$ and the OT plan $\pi^\star$ is typically close to a permutation matrix.

## A.2   LEMMA 1: LOCAL DIRECTIONAL ALIGNMENT OF GW AND DIRICHLET GRADIENTS

We first show that, in a neighbourhood where the consensus geometry is approximately respected, minimizing the GW discrepancy encourages embeddings with low Dirichlet energy on the consensus graph, linking GW alignment to smoothness of $\mathbf{Z}$ with respect to the consensus geometry.

**Lemma 1** (Local directional alignment of GW and Dirichlet gradients). *Under the setup of Eq. equation 13 and $E_{\mathrm{Dir}}(\mathbf{Z}) = \mathrm{tr}(\mathbf{Z}^\top \mathbf{L}^* \mathbf{Z})$ (Chung, 1996). Let*

$$G(\mathbf{Z}) = \nabla_{\mathbf{Z}} d_{\mathrm{GW}}^2(\mathbf{K}_{\mathbf{z}}, \mathbf{C}_{\mathbf{x}}^*), \qquad H(\mathbf{Z}) = \nabla_{\mathbf{Z}} E_{\mathrm{Dir}}(\mathbf{Z}) = 2\mathbf{L}^* \mathbf{Z}.$$

*Assume that:*

1. *$\mathbf{K}_{\mathbf{z}} \to \mathbf{C}_{\mathbf{x}}^*$ and $\pi^\star(\mathbf{Z}) \to \Pi$ as $\mathbf{Z} \to \bar{\mathbf{Z}}$, for a permutation $\Pi$ and the reference embedding $\bar{\mathbf{Z}}$, which is the embedding to exactly represent structural consensus;*

2. *$G$ and $H$ are nonzero at $\bar{\mathbf{Z}}$ and positively colinear, i.e. $G(\bar{\mathbf{Z}}) = \lambda H(\bar{\mathbf{Z}})$ for some $\lambda > 0$.*

*Then for every $\varepsilon > 0$ there exists $\eta > 0$ such that whenever*

$$\|\mathbf{K}_{\mathbf{z}} - \mathbf{C}_{\mathbf{x}}^*\|_F + \|\pi^\star(\mathbf{Z}) - \Pi\|_F < \eta, \quad G(\mathbf{Z}) \neq 0, \ H(\mathbf{Z}) \neq 0,$$

*we have the directional approximation*

$$\left\| \frac{G(\mathbf{Z})}{\|G(\mathbf{Z})\|_F} - \frac{H(\mathbf{Z})}{\|H(\mathbf{Z})\|_F} \right\|_F \leq \varepsilon.$$

*That is, in a small neighbourhood of the reference configuration $\bar{\mathbf{Z}}$, the GW gradient and the Dirichlet gradient point in almost the same direction.*

*Proof.* Under the assumptions on $k$ and the squared loss, the GW objective can be written as a smooth function of the similarity matrix $\mathbf{K_z}$ and the transport plan $\pi^\star(\mathbf{Z})$ (?):

$$d_{\mathrm{GW}}^2(\mathbf{K_z}, \mathbf{C_x^*}) = \sum_{i,j,k,\ell} \left(\mathbf{K_z}(i,j) - \mathbf{C_x^*}(k,\ell)\right)^2 \pi_{ik}^\star(\mathbf{Z})\, \pi_{j\ell}^\star(\mathbf{Z}).$$

Each entry $\mathbf{K_z}(i,j) = k(\|\mathbf{z}_i - \mathbf{z}_j\|^2)$ is a function of $\mathbf{Z}$, and the expression above is a finite sum of smooth functions of $(\mathbf{K_z}, \pi^\star(\mathbf{Z}))$. Hence $G(\mathbf{Z})$ is continuous in a neighbourhood of $\bar{\mathbf{Z}}$. Likewise, $H(\mathbf{Z}) = 2\mathbf{L}^*\mathbf{Z}$ is linear in $\mathbf{Z}$ and therefore continuous.

On the set where $G(\mathbf{Z}) \neq 0$ and $H(\mathbf{Z}) \neq 0$, the normalized gradients

$$u(\mathbf{Z}) := \frac{G(\mathbf{Z})}{\|G(\mathbf{Z})\|_F}, \qquad v(\mathbf{Z}) := \frac{H(\mathbf{Z})}{\|H(\mathbf{Z})\|_F}$$

are continuous functions of $\mathbf{Z}$. By the colinearity assumption, $u(\bar{\mathbf{Z}}) = v(\bar{\mathbf{Z}})$. By continuity of $u$ and $v$, the standard $\varepsilon$–$\delta$ argument implies that for every $\varepsilon > 0$, there exists $\eta > 0$ such that $\|\mathbf{K_z} - \mathbf{C_x^*}\|_F + \|\pi^\star(\mathbf{Z}) - \Pi\|_F < \eta$ entails $\|u(\mathbf{Z}) - v(\mathbf{Z})\|_F \leq \varepsilon$. This is precisely the claimed inequality. $\qquad\square$

Lemma 1 formalizes the statement that, in a near-alignment regime, the GW term pushes $\mathbf{Z}$ in almost the same direction as the Dirichlet energy: infinitesimal gradient steps for the GW loss act like graph smoothing on the consensus geometry.

Intuitively, the Dirichlet energy (Belkin & Niyogi, 2003)

$$E_{\mathrm{Dir}}(\mathbf{Z}) = \frac{1}{2} \sum_{i,j} \mathbf{L}^*(i.j)\|\mathbf{z}_i - \mathbf{z}_j\|^2$$

measures the total "elastic tension" of a spring network with edge weights $a_{ij}$. Minimizing GW distance drives $\mathbf{K_z}$ to match $\mathbf{C_x^*}$, i.e. to embed this graph faithfully. Once this is achieved, no Laplacian-type perturbation can further reduce the tension without breaking the matched structure, which is the content of Lemma 1.

### A.3 LEMMA 2: DIRICHLET ENERGY AND LOW-FREQUENCY STRUCTURE

We now recall a standard spectral decomposition of the Dirichlet energy, which makes explicit that minimizing $E_{\mathrm{Dir}}$ places most of the "mass" of $\mathbf{Z}$ on the low-frequency eigenvectors of the consensus Laplacian.

**Lemma 2** (Spectral decomposition and low-frequency bias)**.** *Let* $\mathbf{L}^* = \mathbf{U}\Lambda\mathbf{U}^\top$ *with* $0 = \lambda_1 \leq \lambda_2 \leq \cdots \leq \lambda_n$*. Define* $\tilde{\mathbf{Z}} = \mathbf{U}^\top\mathbf{Z}$*. Under a norm constraint* $\|\mathbf{Z}\|_F^2 = c$*, one has*

$$E_{\mathrm{Dir}}(\mathbf{Z}) = \mathrm{tr}(\mathbf{Z}^\top\mathbf{L}^*\mathbf{Z}) = \sum_{\ell=1}^n \lambda_\ell\|\tilde{\mathbf{Z}}_{\ell,:}\|_2^2.$$

*Minimizers therefore place maximal energy on the eigenspaces corresponding to the smallest eigenvalues, i.e. on the low-frequency modes of the consensus geometry.*

*Proof.* Using $\mathbf{L}^* = \mathbf{U}\Lambda\mathbf{U}^\top$ and $\tilde{\mathbf{Z}} = \mathbf{U}^\top\mathbf{Z}$,

$$E_{\mathrm{Dir}}(\mathbf{Z}) = \mathrm{tr}\left(\mathbf{Z}^\top\mathbf{U}\Lambda\mathbf{U}^\top\mathbf{Z}\right) = \mathrm{tr}\left(\tilde{\mathbf{Z}}^\top\Lambda\tilde{\mathbf{Z}}\right) = \sum_{\ell=1}^n \lambda_\ell\|\tilde{\mathbf{Z}}_{\ell,:}\|_2^2.$$

The Frobenius norm constraint reads

$$\|\mathbf{Z}\|_F^2 = \mathrm{tr}(\mathbf{Z}^\top\mathbf{Z}) = \mathrm{tr}(\tilde{\mathbf{Z}}^\top\tilde{\mathbf{Z}}) = \sum_{\ell=1}^n \|\tilde{\mathbf{Z}}_{\ell,:}\|_2^2 = c.$$

Thus we minimize a weighted sum $\sum_\ell \lambda_\ell a_\ell$ subject to $\sum_\ell a_\ell = c$ with $a_\ell = \|\tilde{\mathbf{Z}}_{\ell,:}\|_2^2 \geq 0$ (von Luxburg, 2007). Since $0 = \lambda_1 \leq \lambda_2 \leq \cdots \leq \lambda_n$, we have

$$E_{\mathrm{Dir}}(\mathbf{Z}) - \lambda_1 c = \sum_{\ell=2}^n (\lambda_\ell - \lambda_1)\, a_\ell \; \geq \; 0,$$

with strict inequality whenever some $a_\ell > 0$ for $\lambda_\ell > \lambda_1$. Hence any minimizer of $E_{\mathrm{Dir}}$ under the norm constraint concentrates as much energy as possible on indices with the smallest eigenvalues, i.e., the low-frequency eigenvectors of $\mathbf{L}^*$. $\qquad\square$

**Interpretation.** Since $\tilde{\mathbf{Z}}_{\ell,:} = \mathbf{U}_\ell^\top \mathbf{Z}$ is the projection of the embedding onto the $\ell$-th eigenvector of $\mathbf{L}^*$, the expression

$$E_{\mathrm{Dir}}(\mathbf{Z}) = \sum_\ell \lambda_\ell \|\tilde{\mathbf{Z}}_{\ell,:}\|^2$$

states that high-frequency components (large $\lambda_\ell$) are heavily penalized. Thus any $\mathbf{Z}$ minimizing Dirichlet energy must align itself with the low-frequency eigenvectors of $\mathbf{L}^*$; equivalently, $\mathbf{Z}$ becomes "most compatible" with the smooth, large-scale geometry encoded by these eigenvectors.

In particular, if $\tilde{\mathbf{Z}}_{\ell,:}$ is large for small $\lambda_\ell$, then the rows of $\mathbf{Z}$ must be close to the eigenvectors $\mathbf{U}_\ell$, meaning the learned embeddings inherit the global structure of $\mathbf{L}^*$. This formalizes why the structural term preserves shared low-frequency structure.

### A.4 MAIN THEOREM: CONSENSUS ALIGNMENT WITHOUT COLLAPSING RICH MODALITIES

We now combine the two lemmas to articulate our main conceptual point: in this idealized setting, aligning a information-rich modality to a consensus geometry via our GW-based structural regularizer does *not* force the encoder to discard its modality-specific (high-frequency) information. Instead, it primarily constrains the shared low-frequency structure.

**Theorem 2** (GW-based consensus alignment preserves modality-specific information). *Consider a rich modality $R$ and a poorer modality $P$ with embeddings $\mathbf{Z}_R, \mathbf{Z}_P$, similarity matrices $\mathbf{K}_R, \mathbf{K}_P$, and barycenter $\mathbf{C}_{\mathbf{x}}^*$. Assume embeddings are trained with a contrastive loss and the GW regularizer $d_{\mathrm{GW}}^2(\mathbf{K}_{\mathbf{z}}, \mathbf{C}_{\mathbf{x}}^*)$, under the norm control $\|\mathbf{Z}\|_F^2 = c$.*

*Then, in any neighbourhood where $\mathbf{K}_{\mathbf{Z}_m} \approx \mathbf{C}_{\mathbf{x}}^*$:*

1. *By Lemma 1, minimizing GW forces $\mathbf{Z}_m$ to descend in (almost) the same direction as the Dirichlet gradient $\mathbf{L}^* \mathbf{Z}_m$, thus enforcing agreement on the low-frequency structure of the consensus Laplacian.*

2. *By Lemma 2, this alignment constrains only the projections of $\mathbf{Z}_m$ onto the low-frequency eigenspaces of $\mathbf{L}^*$; all components in high-frequency eigenspaces ($\lambda_\ell$ large) remain weakly constrained by the structural term.*

3. *The contrastive objective acts primarily on shared structure and, as observed in multimodal representation learning, does not by itself eliminate modality-specific information: shared information is aligned, while modality-specific details are naturally retained.*

*Consequently, the GW regularizer enforces a* consensus low-frequency geometry *without collapsing the rich modality to the poor one. Modality-specific (high-frequency) information in $\mathbf{Z}_R$ is preserved and remains available for contrastive discrimination, while only the shared geometric structure is aligned.*

### A.5 EMPIRICAL GW DISTANCE

**Theorem 1** (Empirical GW Distance). *Let the kernel matrices $\mathbf{K}_{\mathbf{x}} \in \mathbb{R}^{I \times I}$ and $\mathbf{K}_{\mathbf{y}} \in \mathbb{R}^{J \times J}$ be the similarity matrices conducted by the samples $\mathbf{x}, \mathbf{y}$ from two mm-spaces $\mathcal{X}, \mathcal{Y}$, the empirical GW distance between the samples is:*

$$\hat{d}_{gw}(\mathbf{K}_{\mathbf{x}}, \mathbf{K}_{\mathbf{y}}) := \max_{\mathbf{T} \in \Pi(\hat{\mathbf{p}}_{\mathbf{x}}, \hat{\mathbf{p}}_{\mathbf{y}})} \mathrm{tr}(\mathbf{K}_{\mathbf{x}}^\top \mathbf{T}^\top \mathbf{K}_{\mathbf{y}} \mathbf{T}),$$

*where $\mathbf{T}$ is the doubly-stochastic matrix to model the transport between the two sets of samples.*

*Proof.* Let $\mathcal{X} = \{\mathbf{x}_i\}_{i=1}^I$ and $\mathcal{Y} = \{\mathbf{y}_j\}_{j=1}^J$ be the two finite mm-spaces with uniform empirical marginals $\hat{\mathbf{p}}_{\mathbf{x}} = \frac{1}{I}\mathbf{1}_I$ and $\hat{\mathbf{p}}_{\mathbf{y}} = \frac{1}{J}\mathbf{1}_J$. Denote their intra-modal similarity matrices by $\mathbf{K}_{\mathbf{x}} \in \mathbb{R}^{I \times I}$

and $\mathbf{K_y} \in \mathbb{R}^{J \times J}$, where $(K_\mathbf{x})_{ii'} = \text{sim}(\mathbf{x}_i, \mathbf{x}_{i'})$ and $(K_\mathbf{y})_{jj'} = \text{sim}(\mathbf{y}_j, \mathbf{y}_{j'})$. A cross-domain soft matching is a coupling

$$\mathbf{T} \in \Pi(\hat{\mathbf{p}}_\mathbf{x}, \hat{\mathbf{p}}_\mathbf{y}) := \left\{ \mathbf{T} \geq 0 \mid \mathbf{T}\mathbf{1}_J = \hat{\mathbf{p}}_\mathbf{x}, \ \mathbf{T}^\top \mathbf{1}_I = \hat{\mathbf{p}}_\mathbf{y} \right\}.$$

The empirical GW distance can be written as the minimum expected squared discrepancy of within-domain relations:

$$\hat{d}_{gw}^2(\mathbf{K_x}, \mathbf{K_y}) = \min_{\mathbf{T} \in \Pi(\hat{\mathbf{p}}_\mathbf{x}, \hat{\mathbf{p}}_\mathbf{y})} \sum_{i,i'} \sum_{j,j'} \left((K_\mathbf{x})_{ii'} - (K_\mathbf{y})_{jj'}\right)^2 \mathbf{T}_{ij} \mathbf{T}_{i'j'}. \tag{14}$$

Expand the square in Eq. 14 and group terms:

$$\sum_{i,i',j,j'} \left((K_\mathbf{x})_{ii'} - (K_\mathbf{y})_{jj'}\right)^2 \mathbf{T}_{ij} \mathbf{T}_{i'j'} = A + B - 2 \sum_{i,i',j,j'} (K_\mathbf{x})_{ii'} (K_\mathbf{y})_{jj'} \mathbf{T}_{ij} \mathbf{T}_{i'j'},$$

where $A, B$ are constants

$$A = \sum_{i,i'} (K_\mathbf{x})_{ii'}^2 \, \hat{\mathbf{p}}_\mathbf{x}(i) \, \hat{\mathbf{p}}_\mathbf{x}(i'), \qquad B = \sum_{j,j'} (K_\mathbf{y})_{jj'}^2 \, \hat{\mathbf{p}}_\mathbf{y}(j) \, \hat{\mathbf{p}}_\mathbf{y}(j').$$

Therefore, minimizing Eq. 14 is equivalent to maximizing the quadratic term

$$\max_{\mathbf{T} \in \Pi(\hat{\mathbf{p}}_\mathbf{x}, \hat{\mathbf{p}}_\mathbf{y})} \sum_{i,i',j,j'} (K_\mathbf{x})_{ii'} (K_\mathbf{y})_{jj'} \mathbf{T}_{ij} \mathbf{T}_{i'j'}.$$

In matrix notation, this becomes the quadratic type objective as is in Thrm. 1

$$\hat{d}_{gw}(\mathbf{K_x}, \mathbf{K_y}) = \max_{\mathbf{T} \in \Pi(\hat{\mathbf{p}}_\mathbf{x}, \hat{\mathbf{p}}_\mathbf{y})} \text{tr}(\mathbf{K_x}^\top \mathbf{T}^\top \mathbf{K_y} \mathbf{T}). \tag{15}$$

Consequently, given an optimal plan $\mathbf{T}^*$ estimated by Alg. 2,

$$\hat{d}_{gw}(\mathbf{K_x}, \mathbf{K_y}) = \text{tr}(\mathbf{K_x}^\top \mathbf{T}^{*\top} \mathbf{K_y} \mathbf{T}^*). \tag{16}$$

$\square$

# B IMPLEMENTATION DETAILS

## B.1 OPTIMAL TRANSPORT PLAN ESTIMATION

---

**Algorithm 2** OTEstimation($\hat{\mathbf{K}}, \mathbf{K}$)

---

**Input:** Kernel matrices $\hat{\mathbf{K}} \in \mathbb{R}^{\hat{N} \times \hat{N}}, \mathbf{K} \in \mathbb{R}^{N \times N}$
**Output:** Optimal transport matrix $\mathbf{T}^*$

Initialize $\mathbf{p} \leftarrow \frac{1}{N}\mathbf{1}_N, \quad \hat{\mathbf{p}} \leftarrow \frac{1}{\hat{N}}\mathbf{1}_{\hat{N}}, \quad \mathbf{T} \leftarrow \hat{\mathbf{p}}\mathbf{p}^\top$
**while** *not converged* **do**
$\quad$ // Apply Network simplex algorithm:
$\quad \hat{\mathbf{T}} \leftarrow \arg\max_{\mathbf{T} \in \Pi(\hat{\mathbf{p}}, \mathbf{p})} \text{tr}(\hat{\mathbf{K}}^\top \mathbf{T}^\top \mathbf{K} \mathbf{T})$
$\quad$ // Line search method to find the minimum:
$\quad a \leftarrow -2\,\text{tr}(\hat{\mathbf{K}}^\top \hat{\mathbf{T}}^\top \mathbf{K} \mathbf{T})$
$\quad b \leftarrow \text{tr}((\hat{\mathbf{K}} \odot \hat{\mathbf{K}})\hat{\mathbf{p}}\mathbf{p}^\top + \hat{\mathbf{p}}\mathbf{p}^\top(\mathbf{K} \odot \mathbf{K})^\top)$
$\quad c \leftarrow -2\left(\text{tr}(\hat{\mathbf{K}}^\top \mathbf{T}^\top \mathbf{K}\hat{\mathbf{T}}) + \text{tr}(\hat{\mathbf{K}}^\top \hat{\mathbf{T}}^\top \mathbf{K}\mathbf{T})\right)$
$\quad$ **if** $a > 0$ **then**
$\quad\quad \tau \leftarrow \min(1, \max(0, -\frac{b+c}{2a}))$
$\quad$ **else**
$\quad\quad \tau \leftarrow \begin{cases} 1, & \text{if } a + b + c < 0, \\ 0, & \text{otherwise.} \end{cases}$
$\quad \mathbf{T} \leftarrow (1-\tau)\mathbf{T} + \tau\hat{\mathbf{T}}$
**return** $\mathbf{T}$

---

Algorithm 2 computes an empirical OT plan $\mathbf{T}$ by solving the quadratic program

$$\max_{\mathbf{T} \in \Pi(\hat{\mathbf{p}}, \mathbf{p})} f(\mathbf{T}) := \mathrm{tr}\big(\hat{\mathbf{K}}^\top \mathbf{T}^\top \mathbf{K}\, \mathbf{T}\big),$$

where $\hat{\mathbf{K}}, \mathbf{K} \in \mathbb{R}^{N \times N}$ are intra-domain similarity (or distance) matrices and $\Pi(\hat{\mathbf{p}}, \mathbf{p}) = \{\mathbf{T} \geq 0 \mid \mathbf{T}\mathbf{1} = \hat{\mathbf{p}}, \mathbf{T}^\top \mathbf{1} = \mathbf{p}\}$ is the transportation polytope (doubly-stochastic when $\hat{\mathbf{p}} = \frac{1}{\hat{N}} \mathbf{1}_{\hat{N}}, \mathbf{p} = \frac{1}{N} \mathbf{1}_N$). Here $\odot$ is the Hadamard product, so $(\hat{\mathbf{K}} \odot \hat{\mathbf{K}})$ and $(\mathbf{K} \odot \mathbf{K})$ are elementwise squares of the corresponding kernels, which makes $b$ compact. We initialize with the independent coupling $\mathbf{T} = \hat{\mathbf{p}}\mathbf{p}^\top$ and iterate a Conditional Gradient (Frank–Wolfe; FW) update.

**Network simplex algorithm.** At each iteration, we linearize $f$ and solve

$$\hat{\mathbf{T}} \in \arg\max_{\mathbf{T} \in \Pi(\hat{\mathbf{p}}, \mathbf{p})} \big\langle \mathbf{T},\, \nabla f(\mathbf{T}) \big\rangle.$$

For $f(\mathbf{T}) = \mathrm{tr}(\hat{\mathbf{K}}^\top \mathbf{T}^\top \mathbf{K}\mathbf{T})$, we use the gradient form

$$\nabla f(\mathbf{T}) = \mathbf{K}\, \mathbf{T}\, \hat{\mathbf{K}} + \mathbf{K}^\top \mathbf{T}\, \hat{\mathbf{K}}^\top,$$

which reduces to $2\,\mathbf{K}\,\mathbf{T}\,\hat{\mathbf{K}}$ when $\mathbf{K}, \hat{\mathbf{K}}$ are symmetric. The oracle is a linear transportation problem. We implement it using a network simplex (Flamary et al., 2021; Bonneel et al., 2011).

**Line search.** Define the search segment $\mathbf{T}(\tau) = (1 - \tau)\mathbf{T} + \tau\hat{\mathbf{T}}, \tau \in [0, 1]$. Substituting $\mathbf{T}(\tau)$ into $f$ yields a univariate quadratic $f(\tau) = a\tau^2 + b\tau + c$ whose coefficients admit closed forms. The code computes $(a, b, c)$ and picks the maximizer on $[0, 1]$: $\tau^\star = \min\big(1, \max(0, -(b + c)/(2a))\big)$ if $a > 0$, otherwise $\tau^\star \in \{0, 1\}$ by comparing endpoints. We then set $\mathbf{T} = \mathbf{T}(\tau^\star)$.

## B.2 GW BARYCENTER ESTIMATION

---

**Algorithm 3** GW Barycenter Estimation (mini-batch)

---

**Input:** Intra-modal similarity matrices $\{\mathbf{K}_\mathbf{x}^{(m)}\}_{m=1}^M$ (batch size $N_m$ per modality with $\min\{N_m\} = N$), modality weights $\{\lambda_m\}_{m=1}^M$ with $\lambda_m \geq 0$, $\sum_m \lambda_m = 1$, uniform marginal $\hat{\mathbf{p}} = \frac{1}{N}\mathbf{1}_N, \mathbf{p}^{(m)} = \frac{1}{N_m}\mathbf{1}_{N_m}$, max iters $T_{\max}$

**Output:** Batch-wise structural consensus (GW barycenter) $\mathbf{C}_\mathbf{x}^* \in \mathbb{R}^{N \times N}$

Initialize $\mathbf{C}_\mathbf{x}$ as the weighted average of $\mathbf{K}_\mathbf{x}^{(m)}$
**for** $t \leftarrow 0$ **to** $T_{\max} - 1$ **do**
    **for** $m \leftarrow 1$ **to** $M$ **do**
        $\mathbf{T}^{(m)} \leftarrow \texttt{OTEstimation}\Big(\mathbf{C}_\mathbf{x}, \mathbf{K}_\mathbf{x}^{(m)}\Big)$ ;              // Alg. 2
    $\widetilde{\mathbf{C}} \leftarrow \sum_{m=1}^M \lambda_m \mathbf{T}^{(m)} \mathbf{K}_\mathbf{x}^{(m)} \mathbf{T}^{(m)\top}$
    $\mathbf{C}_\mathbf{x} \leftarrow \widetilde{\mathbf{C}} \oslash \Big(\hat{\mathbf{p}}\, \mathbf{p}^{(m)\top}\Big)$
**return** $\mathbf{C}_\mathbf{x}^* \leftarrow \mathbf{C}_\mathbf{x}$

---

Consider the barycenter objective (Def. 2):

$$\mathbf{C}_\mathbf{x}^* = \arg\min_{\mathbf{C}_\mathbf{x} \in \mathcal{M}} \sum_{m=1}^M \lambda_m \cdot d_{gw}(\mathbf{C}_\mathbf{x}, \mathbf{K}_\mathbf{x}^{(m)}), \qquad \lambda_m \geq 0,\ \sum_{m=1}^M \lambda_m = 1.$$

According to the discrete empirical GW distance form (Thrm. 1), each term differs from a constant by a (negative) maximized trace. Fix couplings $\{\mathbf{T}^{(m)}\}_{m=1}^M$ with $\mathbf{T}^{(m)} \in \Pi(\hat{\mathbf{p}}, \mathbf{p}^{(m)})$ for the current consensus $\mathbf{C}_\mathbf{x}$, and define

$$\mathbf{A}^{(m)} := \mathbf{T}^{(m)} \mathbf{K}_\mathbf{x}^{(m)} \mathbf{T}^{(m)^\top} \in \mathbb{R}^{N \times N}.$$

as $\mathbf{C_x}$-independent constants, the objective reduces to

$$\mathcal{J}(\mathbf{C_x}) = -2 \sum_{m=1}^{M} \lambda_m \operatorname{tr}\big(\mathbf{C_x}^{\top} \mathbf{A}^{(m)}\big).$$

Following the standard GW-barycenter normalization (as in Eq. (8) of Gong et al. (2022)), we take the derivative with respect to $\mathbf{C}$ and set it to zero

$$\frac{\partial \mathcal{J}(\mathbf{C_x})}{\partial \mathbf{C_x}} = \mathbf{0} \quad \Rightarrow \quad \mathbf{C_x} = \Big( \sum_{m=1}^{M} \lambda_m \mathbf{A}^{(m)} \Big) \oslash \big( \hat{\mathbf{p}} \, \hat{\mathbf{p}}^{\top} \big),$$

i.e.

$$\mathbf{C_x} \leftarrow \widetilde{\mathbf{C}} \oslash \big( \hat{\mathbf{p}} \, \hat{\mathbf{p}}^{\top} \big), \quad \widetilde{\mathbf{C}} = \sum_{m=1}^{M} \lambda_m \, \mathbf{T}^{(m)} \, \mathbf{K_x}^{(m)} \, \mathbf{T}^{(m)}{}^{\top}. \tag{17}$$

Here $\oslash$ denotes the element-wise division.

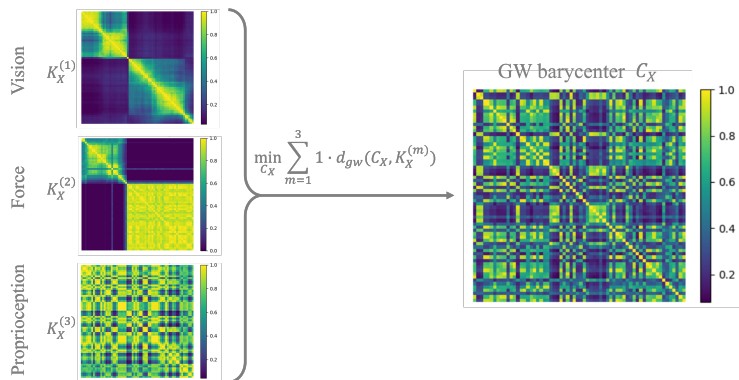

Figure 4: GW barycenter of input–space kernels on the VFP dataset. Left: intra–modal similarity matrices $K_{\mathbf{x}}^{(1)}$ (Vision), $K_{\mathbf{x}}^{(2)}$ (Force), and $K_{\mathbf{x}}^{(3)}$ (Proprioception), each min–max normalized for display. Right: the batch-wise structural consensus $\mathbf{C_x}^{*}$ obtained by solving $\min_{\mathbf{C_x}} \sum_{m=1}^{3} \lambda_m \, d_{gw}\big(\mathbf{C_x}, K_{\mathbf{x}}^{(m)}\big)$ (with $\lambda_m{=}1$ here). The barycenter preserves recurrent block/trajectory patterns shared across modalities while smoothing modality-specific artifacts, and is later used to regularize the embedding-space geometry in Stage 2. The batch size is 64.

### B.3 IMPLEMENTATION DETAILS

**Implementation: Time-Series Cluster Kernel** We use the Time-series Cluster Kernel (TCK; Mikalsen et al. (2018)) to build intra-modal similarity matrices for time-series modalities (e.g., force/torque). TCK fits an ensemble of diagonal covariance Gaussian mixture models (GMMs) with informative priors and computes a posterior membership vector per sample

$$\mathbf{\Pi}_i(q) = \big(\pi_1^{(i)}(q), \dots, \pi_{G_q}^{(i)}(q)\big)^{\top}, \qquad \sum_{g=1}^{G_q} \pi_g^{(i)}(q) = 1,$$

where each component $\pi_g^{(i)}(q)$ is the posterior responsibility of mixture $g$ for sequence $i$ under the $q$-th GMM, i.e.

$$\pi_g^{(i)}(q) = p\Big(z{=}g \,\Big|\, \mathbf{x}_i^{(q)}; \hat{\theta}_q\Big),$$

where $z$ is the latent mixture index, $\hat{\theta}_q$ is the MAP-EM estimate of the $q$-th model parameters, and $\mathbf{x}_i^{(q)}$ is the subsequence of $i$ restricted to the time window and variable subset chosen by that ensemble member. The final kernel is the sum of posterior inner products over the ensemble:

$$(K_{\text{TCK}})_{ij} \leftarrow \sum_{q \in \mathcal{Q}} \mathbf{\Pi}_i(q)^{\top} \mathbf{\Pi}_j(q),$$

which is positive semidefinite as a sum of linear kernels. In practice, for time–critical training, we precompute the full TCK matrix for the entire force dataset (about $10^5$ sequences) to get a single symmetric matrix $\mathbf{K}_{\text{TCK}} \in \mathbb{R}^{N \times N}$. During mini-batch training, the intra-modal similarity submatrix for an index set $\mathcal{I} \subseteq \{1, \dots, N\}$ is obtained by simple indexing

$$\mathbf{K}_{\text{batch}} \;=\; K_{\text{TCK}}[\mathcal{I}, \mathcal{I}],$$

thus avoiding repeated TCK fits inside the inner learning loop. We follow the original TCK protocol to induce ensemble diversity (random time windows and variable subsets, random initializations, and varying mixture counts), and we cache per-member posteriors to enable fast posterior lookups at test time. See §4.1–4.4 of Mikalsen et al. (2018) for modeling details. In practice, we set the maximal number of mixtures $C$ and the number of randomizations $Q$ as the only user-set hyperparameters. We set $C=30$ and $Q=15$ for force/torque signals in the VFP and VFD settings.

**Implementation: Pre-trained features**   We use pre-trained feature extractors for some modalities to produce modality-specific features whose pairwise similarities form the input-space kernels $\{\mathbf{K}_{\mathbf{x}}^{(m)}\}_{m=1}^M$ used by our structural consensus $\mathbf{C}_{\mathbf{x}}^*$. For the time-series modality (e.g., force/torque), we directly use the TCK method to obtain the input-space kernels.

*Pre-trained feature extractors (frozen).*

- **Vision / Depth / Tactile / Optical Flow:** Vision Transformer (ViT-B/16; **?**) via `timm` (**?**), taking the final [CLS] embedding. Single-channel inputs (e.g., depth) are replicated to 3 channels before preprocessing.

- **Force:** Time-Series Cluster Kernel (TCK; Mikalsen et al. 2018) directly forms $\mathbf{K}_{\mathbf{x}}^{(\text{force})}$ (Sec. B.3).

- **Audio (VAT):** A frozen Audio Spectrogram Transformer (AST-B, AudioSet-pretrained; **?**) on log-mel spectrograms; we take the [CLS] embedding and build $\mathbf{K}_{\mathbf{x}}^{(\text{aud})}$ with a simple similarity (cosine or RBF).

- **Other modalities:** RBF kernel on frozen features.

**Implementation: Modality Encoders**   To avoid architectural confounds, all methods share identical backbones and training schedules. In UniOMA (Stage 2), each modality encoder $\mathcal{E}_\theta^{(m)}$ produces a feature $\mathbf{h}^{(m)} \in \mathbb{R}^{d_h}$, which is passed through a *modality-specific MLP projector* $g_\theta^{(m)}$ to a *shared* embedding size $d=256$:

$$\mathbf{z}^{(m)} \;=\; g_\theta^{(m)}\big(\mathcal{E}_\theta^{(m)}(\mathbf{x}^{(m)})\big) \in \mathbb{R}^d, \quad f_\theta^{(m)} = g_\theta^{(m)} \cdot \mathcal{E}_\theta^{(m)}$$

and the embedding-space kernel within a mini-batch is

$$\big(\mathbf{K}_{\mathbf{z}}^{(m)}\big)_{ij} = \exp\Big(-\gamma \, \|\mathbf{z}_i^{(m)} - \mathbf{z}_j^{(m)}\|_2^2\Big), \qquad \gamma \;=\; \frac{20}{d} \,,$$

unless stated otherwise. (Stage 1 input-space kernels $\{\mathbf{K}_{\mathbf{x}}^{(m)}\}$ are computed independently using frozen extractors; see Sec. B.3.)

*Backbones.*

- **Vision / Depth / Tactile (image-based).** A 2D CNN (ResNet-18). Single-channel inputs (e.g., depth, some tactile images) are replicated to 3 channels before feeding into the backbone.

- **Force (time series).** A 1D temporal ConvNet built from stacked causal Conv1D layers (kernel size 2, stride 2) with LeakyReLU activations; the final feature map is flattened to obtain a fixed-length embedding.

- **Proprioception.** A 3-layer MLP with LeakyReLU activations, mapping the low-dimensional pose / joint vector to the shared embedding space.

- **Audio (VAT).** A 1D CNN with three convolutional blocks (channels $1 \rightarrow 64 \rightarrow 128 \rightarrow 256$, kernel size 5, stride 2), each followed by ReLU, then `AdaptiveAvgPool1d(1)`, `flatten()`, and a final `Linear(256 \rightarrow d_h)`.

- **Action.** A small MLP mapping the action vector to a compact embedding, implemented as `Linear`$(d_a \to 32)$–LeakyReLU–`Linear`$(32 \to 32)$–LeakyReLU, matching the code in `ActionEncoder`.
- **Contact.** A lightweight MLP applied to the binary contact state, using a structure analogous to the action branch (two `Linear` layers with LeakyReLU) to obtain a 32-dimensional embedding.
- **Optical Flow.** We take one channel of the dense optical-flow field (e.g., horizontal component or magnitude), resize it to $128 \times 128$, replicate it to 3 channels, and feed it through the same ResNet-18 image encoder as for RGB and depth.

We fix the projector output to $d = 256$, use the same temperature $\tau$ for the contrastive term, and share optimizer, batch size, and schedule across methods. UniOMA augments the contrastive loss with a GW-barycenter regularizer (weight $\alpha$) and learnable modality weights $\{\lambda_m\}$ (softmax-parameterized to enforce $\lambda_m \geq 0$ and $\sum_m \lambda_m = 1$). Encoders and projectors are trained end-to-end with the UniOMA objective; the structure-aware term is computed on $\{\mathbf{K}_{\mathbf{z}}^{(m)}\}$, while Stage 1 kernels $\{\mathbf{K}_{\mathbf{x}}^{(m)}\}$ remain fixed within each epoch.

**Implementation: Hyper-parameters** *Shared training.* Unless otherwise noted, all methods use the same backbone–projector settings. We optimize with AdamW (learning rate $3 \times 10^{-4}$, weight decay $10^{-4}$, $\beta_1 = 0.9$, $\beta_2 = 0.999$), batch size 64, and temperature $\tau = 0.1$. Each modality head outputs a $d = 256$-dimensional embedding via a lightweight MLP projector (shared width across modalities). We train for 200 epochs with early stopping on the validation metric when applicable, and report mean$\pm$std over 10 independent seeds.

*Stage-1 input-space kernels.* Pre-trained feature extractors for vision/depth/tactile (ViT-B/16 via `timm`) are *frozen* to compute $\{\mathbf{K}_{\mathbf{x}}^{(m)}\}$. For force/torque we use TCK with max mixtures $C = 30$ and randomizations $Q = 15$ following §B.3. For VAT audio, we use AST-B as in Sec. B.3 to form features and then an RBF kernel. To avoid repeated online estimation during Stage 2, we compute force's full dataset kernel once and cache it; mini-batch kernels $\mathbf{K}_{\text{batch}}^{(\text{force})}$ are obtained by submatrix indexing.

*Stage-2 embedding-space kernels.* All modalities use the same Gaussian kernel

$$\left(\mathbf{K}_{\mathbf{z}}^{(m)}\right)_{ij} = \exp\left(-\gamma \|\mathbf{z}_i^{(m)} - \mathbf{z}_j^{(m)}\|_2^2\right),$$

with a shared, modality-invariant scale $\gamma = 20/d, d = 256$.

*UniOMA-specific.* The GW regularization weight $\alpha = 1000$. Modality weights $\{\lambda_m\}$ are learnable with a softmax parameterization ($\lambda_m \geq 0$, $\sum_m \lambda_m = 1$) and initialized uniformly. For the coupling oracle in `OTEstimation` we use a Frank–Wolfe linearization; the linear subproblem is solved with a network-simplex transportation solver. The line search on the FW segment uses the closed-form quadratic coefficients $(a, b, c)$ derived in Appx. B.1. GW barycenter iterations are run with a maximum of $T_{\max} = 5$ per inner-loop (in §D.1 we analyze the solidity of this choice).

To further justify these design choices, we provide a hyper-parameter ablation in Appx. **??**, where we evaluate the effects of $\gamma$, $\alpha$, $T_{\max}$, and alternative kernel choices.

## C  DATASETS AND PREPROCESSING

We detail the exact splitting, windowing, and per–modality preprocessing used in our experiments. Unless specified, all randomization uses a fixed seed (`seed=42`), and splits are performed at the file/trajectory level to avoid leakage.

**VFD / VFP (Vision–Force–Depth / Vision–Force–Proprioception).** We use `test_ratio` = 0.2 at the file level with `seed` = 42 (train vs. test); validation set shares the test set. Each episode has a length 32. For time step $t$, we form a fixed history window of length $L$ for force (default $L = 32$) and read targets at $t+1$. RGB images are center–cropped to $128 \times 128$, normalized by ImageNet statistics (mean $[0.485, 0.456, 0.406]$, std $[0.229, 0.224, 0.225]$). Depth is stored as $(128, 128, 1)$, normalized by mean 0.5/std 0.5, and used as single–channel tensors. Force–torque histories are

truncated to the last $L$ steps. The resulting tensor has shape $[b \times L \times 6]$. Proprioception is parsed from the first 7 pose components (end effector position/orientation) in the loader and returned as $[b \times 7]$ at the current step.

*Tasks.* For **VFD**, we follow the main text: (1) next–step end–effector orientation regression (4D), using $(\text{RGB}_t, \text{F/T}_{t-L+1:t}, \text{Depth}_t)$ as inputs; (2) modality–consistency discrimination with negatives produced by cross–time/trajectory shuffles at 50/50 balance. For **VFP**, we perform next–step contact prediction (binary) using $(\text{RGB}_t, \text{F/T}_{t-L+1:t}, \text{Proprio}_t)$; class balance is enforced by uniform sampling across trajectories.

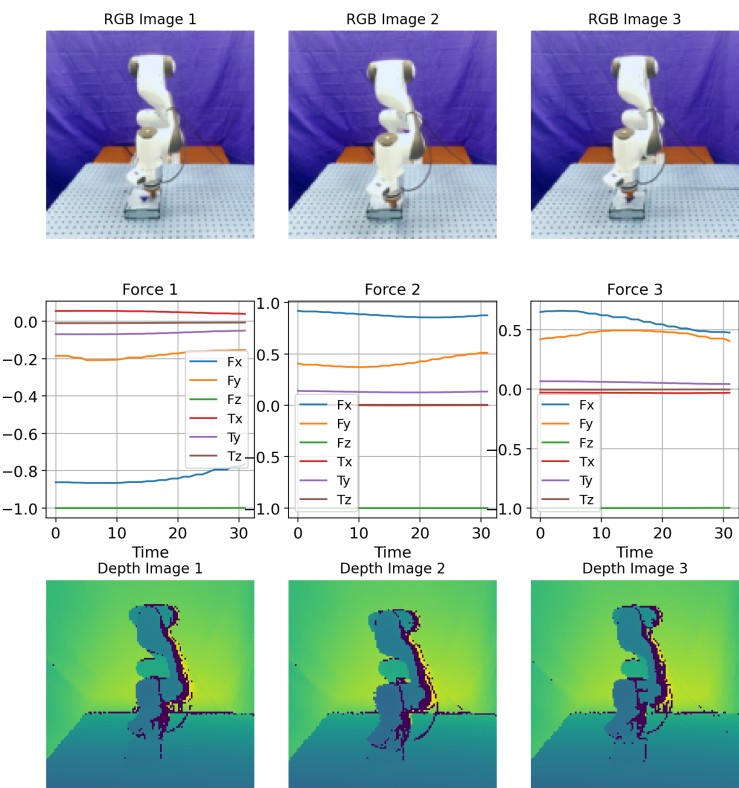

Figure 5: **VFD overview.** Synchronized windows of RGB images, force–torque signals (last $L{=}32$ steps), and depth camera images. Images are center–cropped to $128 \times 128$ and normalized. Depth images are normalized with mean/std $0.5$.

**MuJoCo Push.** A planar pushing task with a Franka Panda arm interacting with a puck. *Image modality:* we use sequences of grayscale frames. Each sample contains a length $S = 32$ subsequence of $32 \times 32$ frames, forming tensors of shape $(B, S, 1, 32, 32)$. Force-torque modality uses the current signal to form tensors of shape $(B, 6)$, and end-effector pose modality forms $(B, 7)$.

**VAT (Vision–Audio–Tactile).** We assemble object–level triplets from per–class folders. We use predefined train/val/test directory structures over a fixed object list. Labels for the retrieval tasks are integer–encoded. Visual and tactile images are resized to $246 \times 246$ and normalized by ImageNet statistics. Audio is loaded at its native sampling rate; at test time, the raw waveform is truncated to `TARGET_LENGTH` $= 132{,}300$ samples. The final shape of the tensors is $(B, 132{,}000)$

*Task.* Cross–modal retrieval with relevance at the object identity level; we report direction–specific MAP on the test set.

**MultiBench 4–7 Modality Scalability.** To evaluate whether UniOMA scales beyond three modalities, we construct an additional downstream classification benchmark using a multi-sensor subset

of the MultiBench dataset. This setting allows us to progressively increase the number of modalities and test whether the learned structural consensus remains stable as modality count grows.

*Modalities.* We select seven heterogeneous sensing streams commonly used in robotic manipulation:

$$\text{Vision (RGB)} : [3{\times}128{\times}128], \qquad \text{Depth} : [1{\times}128{\times}128],$$
$$\text{Force–Torque} : [6], \quad \text{Proprioception} : [7], \quad \text{Action} : [d_a],$$
$$\text{Contact state} : [1], \quad \text{Optical Flow} : [2{\times}128{\times}128].$$

For a given experiment with $M \in \{4, 5, 6, 7\}$ modalities, we take the first $M$ modalities from this list. All modalities are independently normalized using training-set statistics following the MultiBench protocol.

*Task: trajectory-consistency classification.* Given two multimodal or single-modal samples, the model must classify whether they originate from the **same trajectory**. Positive pairs are sampled from two timesteps of the same trajectory; negatives are sampled across distinct trajectories. This task directly evaluates whether embeddings preserve the trajectory-level structure across multiple modalities.

# D ADDITIONAL EXPERIMENTS

## D.1 HYPER-PARAMETER ANALYSIS

**RBF kernel scale $\gamma$.** We use an RBF kernel in the embedding space:

$$\big(\mathbf{K}_{\mathbf{z}}^{(m)}\big)_{ij} = \exp\big(-\gamma_m \,\|\mathbf{z}_i^{(m)} - \mathbf{z}_j^{(m)}\|_2^2\big).$$

Because distance scales differ by modality, we set $\gamma_m$ per modality based on empirical pairwise distances at convergence: $\gamma_{\text{vision/depth/tactile}} = 5$, $\gamma_{\text{proprio}} = 20$, and $\gamma = 10$ for other learnable streams unless stated. Performance is stable within a $\times 0.5 \sim \times 2$ range; very small $\gamma$ over-smooths similarities, while very large $\gamma$ over-peaks them.

**Number of GW barycenter iterations $T_{\max}$.** Let $\mathbf{C}^{(t)}$ be the consensus at inner-loop iteration $t$ in Alg. 3. We monitor the relative Frobenius change $\Delta_t = \|\mathbf{C}^{(t)} - \mathbf{C}^{(t-1)}\|_F / \|\mathbf{C}^{(t-1)}\|_F$ and the trace objective $\sum_m \lambda_m \operatorname{tr}\big(\mathbf{C}^{(t)\top} \mathbf{T}^{(m)\top} \mathbf{K}_{\mathbf{x}}^{(m)} \mathbf{T}^{(m)}\big)$. Both stabilize rapidly; after $t{=}5$ further changes are negligible ($\Delta_t < 10^{-3}$). We therefore fix $T_{\max} = 5$ for all reported results.

**Ablation: hyper-parameter effects.** We empirically ablate three key hyper-parameters of UniOMA—the RBF kernel scale $\gamma$, the GW regularization weight $\lambda$, and the number of barycenter iterations $T_{\max}$ as well as the choice of graph-based kernels (RBF, Laplacian affinity, and UMAP fuzzy simplicial set). Table 3 summarizes results on the VFD classification task (Top-1 accuracy, mean $\pm$ std over 10 seeds). Performance is stable across a broad range of values around our default settings; extremely small or large $\gamma$ mildly hurts performance by over-smoothing or over-peaking similarities, while too small $\lambda$ under-utilizes structural alignment and too large $\lambda$ marginally over-regularizes the embeddings. The solver iteration number $T_{\max}$ shows a clear plateau around $T_{\max} = 5$, confirming that a small number of GW barycenter iterations is sufficient in practice. Finally, replacing the RBF kernel with Laplacian or UMAP-based kernels yields comparable or slightly lower accuracy, supporting RBF as a simple and competitive default.

## D.2 ADDITIONAL QUALITATIVE VISUALIZATIONS

For the benefit of the reader, we provide further qualitative visualizations of the structural alignment induced by UniOMA. Figure 7 summarizes four aspects on the 7-modality benchmark: (a) input-space similarity matrices for each modality; (b) the corresponding batch-wise GW barycenter; (c) a t-SNE embedding of the learned representations; and (d) the learned modality weights. Together, these views illustrate how UniOMA preserves shared structure while remaining interpretable.

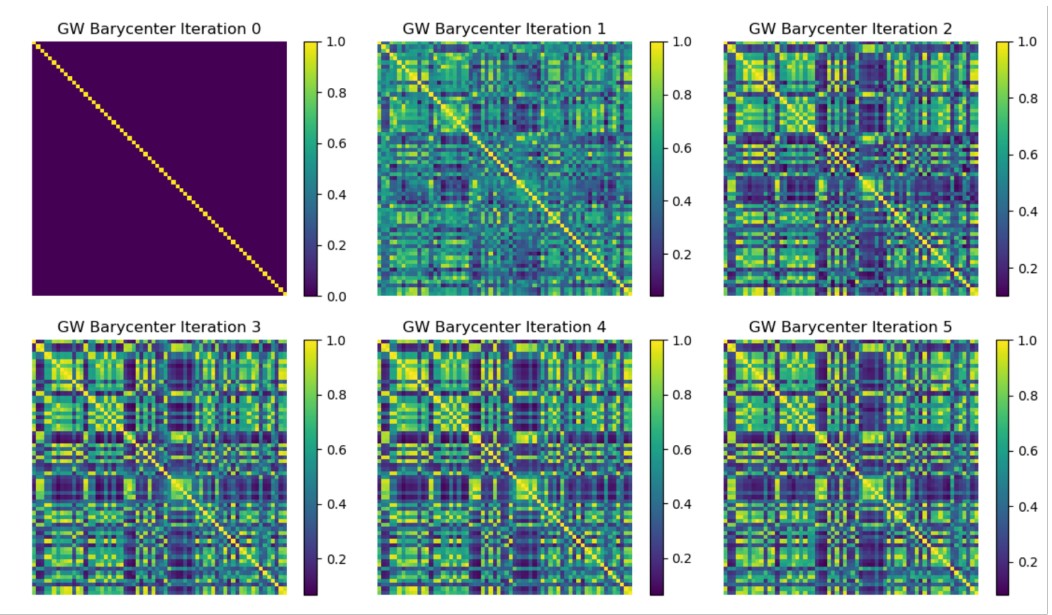

Figure 6: GW barycenter inner-loop: structural consensus across iterations $t$ in Alg. 3. By $t{=}5$, both geometry and objective are effectively stable, thus we choose $T_{\max}{=}5$ and the batch size is 64.

Table 3: Ablation of UniOMA hyper-parameters and graph-based kernels on the VFD classification task (Top-1 accuracy, %, mean $\pm$ std over 10 seeds). We vary the RBF kernel scale $\gamma$ of the image modality, GW weight $\lambda$, and the number of barycenter iterations $T_{\max}$ around the default settings, and compare to median-rule RBF kernel. UniOMA is robust across a wide range of values; our default choices (in **bold**) lie near the center of each stable regime.

| **RBF kernel scale $\gamma$** | |
| --- | --- |
| $\gamma = 1$ | $91.87 \pm 0.05$ |
| $\gamma = 2$ | $92.15 \pm 0.04$ |
| $\gamma = 5$ | $92.44 \pm 0.02$ |
| $\gamma = 10$ | $92.42 \pm 0.02$ |
| $\gamma = 20$ | $92.10 \pm 0.03$ |
| **GW weight $\lambda$** | |
| $\lambda = 200$ | $92.12 \pm 0.02$ |
| $\lambda = 500$ | $92.32 \pm 0.02$ |
| $\lambda = 1000$ | $92.44 \pm 0.02$ |
| $\lambda = 2000$ | $92.40 \pm 0.04$ |
| $\lambda = 5000$ | $92.28 \pm 0.05$ |
| **Barycenter iterations $T_{\max}$** | |
| $T_{\max} = 3$ | $92.05 \pm 0.03$ |
| $T_{\max} = 4$ | $92.27 \pm 0.03$ |
| $T_{\max} = 5$ | $92.44 \pm 0.02$ |
| $T_{\max} = 6$ | $92.45 \pm 0.02$ |
| $T_{\max} = 7$ | $92.44 \pm 0.02$ |
| **Adaptive kernel choice** | |
| RBF (median-rule) | $92.53 \pm 0.03$ |

# E LLM USAGE STATEMENT

This work does not incorporate large language models (LLMs) as a key, novel, or unconventional component of the method, experiments, or analysis. Any LLM assistance was limited to the writ-

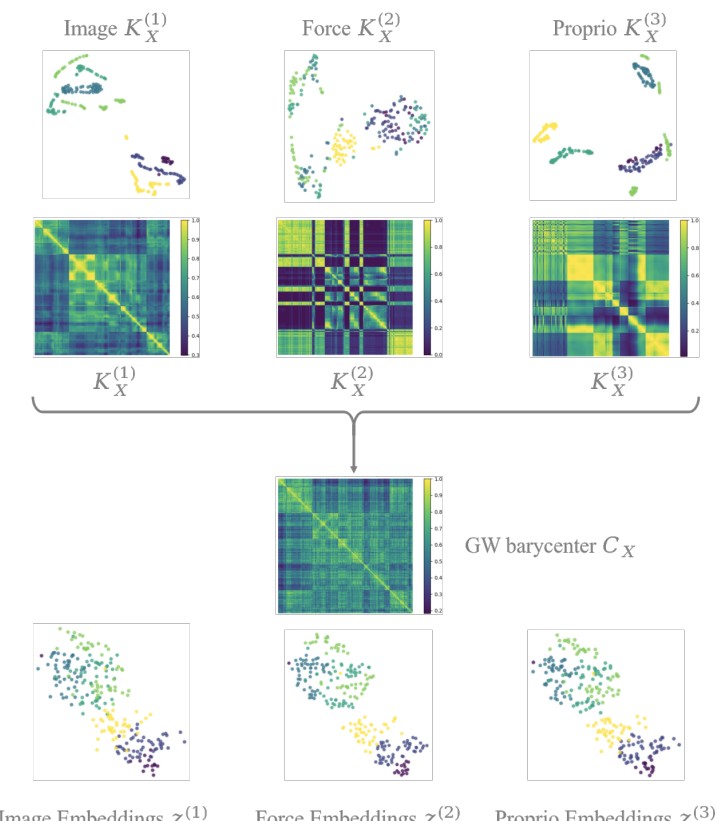

Figure 7: Qualitative visualizations of UniOMA on the VTP classification benchmark. We choose a mini-batch with size 256 to illustrate the interpretation of the GW barycenter and the aligned embeddings. **First row:** t-SNE visualizations of the input modalities (vision, force, and proprioception), showing clear sub-cluster structures indicating different trajectories (6 colors of the points indicating 6 trajectories). **Second row:** Input-space similarity matrices for vision, force, and depth, showing trajectory-wise block structure and modality-specific artifacts. **Middle-bottom:** The batch-wise GW barycenter $C_{\mathbf{x}}^*$, which preserves the shared block structure while smoothing modality-specific noise. **Last row:** t-SNE of the learned embeddings, where trajectories form coherent clusters across modalities, indicating successful structural alignment beyond pairwise correspondence.

ing refinement (grammar, clarity, and copy-editing). All technical formulation, algorithms, proofs, hyperparameters, implementations, and results were created and validated by the authors.

