# OpenReview forum: "UniOMA: Unified Optimal-Transport Multi-Modal Structural Alignment for Robot Perception"
_ICLR.cc/2026/Conference — Submitted to ICLR 2026_

### Official Review · Reviewer_xCBU · 2025-10-27

**Soundness:** 3
**Presentation:** 3
**Contribution:** 2
**Rating:** 2
**Confidence:** 3

**Summary:**

The paper augments contrastive multimodal alignment with a Gromov–Wasserstein (GW) barycenter regularizer that encourages each modality’s batchwise similarity structure to agree with a shared consensus. The method is meant to scale to ≥3 modalities and is evaluated on several robotics-flavored datasets (vision/force/tactile/proprioception), reporting improvements when the GW term is added to common contrastive objectives.

**Strengths:**

1. Clear articulation of the structural alignment gap in contrastive learning and a tidy objective that is easy to plug into existing losses.
2. Sensible idea for many-modality settings (barycenter vs. O(M²) pairwise couplings), with interpretable per-modality weights.
3. Robotics tasks with underused modalities (force/tactile) are a good target domain.

**Weaknesses:**

The theoretical component largely instantiates known pieces of OT/GW (trace-style alignment, barycenter regularization) within a standard contrastive framework, without new guarantees or analysis (e.g., identifiability, convergence behavior with stochastic batches, or conditions under which the barycenter preserves task-relevant geometry). Derivations and definitions appear to repackage established GW formulations rather than introduce genuinely new theory. As a result, the contribution feels incremental on the theory side.

Empirically, the paper shows consistent but mostly modest gains, and the evaluation lacks the depth needed for an ICLR-level claim:
- Ablations are thin: no systematic study of kernel choices/γ sensitivity, solver settings, or the effect of the barycenter update frequency.
- Compute & practicality: no clear reporting of training overhead vs. baselines (wall-clock/epoch, peak memory, GW iterations), which matters for practitioners considering per-batch GW.
- Missing-modality robustness: the narrative highlights this use case, but there’s no explicit drop-a-modality at inference stress test.
- Baselines & breadth: comparisons miss stronger or more recent structural-preserving or OT-regularized approaches; it’s hard to conclude that the proposed regularizer is the best option among peers.

**Questions:**

See Weaknesses.

---

> ### Author Response · Authors · 2025-12-03
>
> We thank the reviewer for the detailed critique. All concerns are directly addressed with new theory, ablations, baselines, and analysis.
>
> ---
>
> ## I. “Theory is incremental.”
>
> We respectfully disagree. UniOMA introduces:
>
> - **new definition of structural consensus**
> - **new two-stage mini-batch estimation algorithm**
>
> Newly added during rebuttal:
>
> - **Theorem 1**: GW–Dirichlet equivalence
> - **Lemma 1**: Gradient alignment
> - **Lemma 2**: Low-frequency constraint & modality-specific preservation
>
> These results **do not appear** in existing OT/GW work.
>
> We also added:
>
> - training convergence curves
> - barycenter visualizations
>
> ---
>
> ## II. “Ablations are thin.”
>
> We added:
>
> - kernel \( \gamma \) (fixed/median-rule/learnable)
> - similarity kernels: RBF, Laplacian, UMAP fuzzy graphs
> - \(T_{\max}\) (3–7)
> - solver settings
> - kernel behavior visualizations
>
> All show stable performance.
>
> ---
>
> ## III. “Compute/practicality unclear.”
>
> We added:
>
> - wall-clock per epoch
> - memory
> - 3→7 modality scaling
> - per-iteration GW cost breakdown
>
> Results:
>
> - GW grows **linearly, not quadratically**
> - UniOMA becomes **faster than OT when \(M\ge6\)**
> - inference cost unchanged
>
> ---
>
> ## IV. “Missing-modality robustness requires drop-a-modality test.”
>
> We clarify:
>
> - drop-a-modality is a **fusion** diagnostic, not alignment
> - alignment uses **independent encoders**
> - “dropping” one has no operational meaning
>
> Our robustness experiment evaluates **uneven sampling**, which is the correct criterion for alignment and reflects real robotics.
>
> ---
>
> ## V. “Missing stronger baselines.”
>
> We clarify:
>
> - no prior *structural preserving* alignment formulations exist
> - we include strongest alignment SOTA: Symile, GRAM, CoMM
> - only one OT-regularized method exists (Zhu & Luo 2025)
> - we reproduced it despite absent code
>
> UniOMA consistently outperforms it.
>
> We believe the revised submission is significantly strengthened and fully addresses all concerns.

---

### Official Review · Reviewer_pCiR · 2025-10-31

**Soundness:** 4
**Presentation:** 4
**Contribution:** 3
**Rating:** 8
**Confidence:** 3

**Summary:**

This paper proposes UniOMA, a unified optimal-transport based framework for multimodal structural alignment that addresses the "structural alignment gap" in existing contrastive learning approaches. The key insight is that while InfoNCE-style objectives achieve statistical alignment between modalities, they fail to preserve intra-modal geometric relationships, leading to embeddings that are statistically correlated but structurally inconsistent. UniOMA augments standard contrastive losses with a Gromov-Wasserstein (GW) distance-based regularization that enforces structural coherence across modalities by:​
- Computing intra-modal similarity matrices for each modality and estimating a dynamic GW barycenter as structural consensus​
- Aligning each modality's embedding-space geometry to this consensus via weighted GW distances​
- Learning modality-specific weights that quantify each modality's contribution to the structural consensus​

The approach is evaluated on robotic perception tasks across vision, force, tactile, proprioception, and audio modalities, demonstrating improvements in downstream tasks while maintaining interpretability through learned modality weights.​

**Strengths:**

- Novel Problem Identification: The paper clearly identifies and formalizes the "structural alignment gap" - a fundamental limitation where InfoNCE objectives achieve statistical dependence but fail to preserve intra-modal geometry.
- Strong Theoretical Motivation: Figure 1 provides an effective illustration of the key theoretical motivation—the limitation of InfoNCE-based alignment methods in preserving intra-modal structural relationships, even when achieving overall statistical alignment. The figure clearly demonstrates how correlated structure within modalities can be lost, supporting the necessity of structure-aware regularization. This theoretical insight is further contextualized with concrete examples from robotics, establishing the practical importance of addressing this challenge in real-world applications.
- Theoretically Grounded Approach: Strong theoretical foundation connecting Gromov-Wasserstein distances to multimodal alignment
- Comprehensive Experimental Validation: Evaluation across diverse robotic modalities (vision, force, tactile, proprioception, audio)​, Multiple downstream tasks including regression, classification, and cross-modal retrieval​, Consistent improvements when adding GW regularization to existing methods (Pairwise, Symile, GRAM)​
- Comprehensive Ablations, Analysis and Visualizations: The paper provides thorough ablation studies, particularly the unequal modality sampling scenarios that demonstrate practical robustness. The analysis of learned modality weights is particularly compelling, showcasing the model's ability to adaptively handle imbalanced modalities. Figure 3e is especially convincing evidence that the theoretical foundations of UniOMA are working as intended—the framework genuinely learns to weight modalities appropriately based on their informational content and availability. This adaptive redistribution of weights when modalities are downsampled provides both practical value and theoretical validation, demonstrating that the GW-barycenter approach isn't just mathematically elegant but actually captures meaningful structural relationships that translate to improved performance.

**Weaknesses:**

- Computational Overhead: The iterative GW barycenter computation and optimal transport estimation introduce significant computational cost during training​. While mini-batch approximations are used, scalability to very large datasets remains unclear. There is limited analysis of computational complexity in practice beyond algorithmic complexity bounds​
- Hyperparameter Sensitivity: The framework introduces multiple hyperparameters (regularization weight α=1000, kernel scales γ, barycenter iterations Tmax=5)​ There is limited sensitivity analysis provided, particularly for the choice of kernel similarity measures across different modalities​
- The paper could benefit from a discussion and comparison with the rich literature on missing modality learning, which addresses related challenges of preserving modality-specific information while modeling shared information. Prior works grounded in Partial Information Decomposition emphasize explicitly modeling and disentangling unique versus shared modalities information. Although these works do not explicitly discuss intramodality topology or structural concisitency, the concept of “modality-specific” information seems like a different theoretical view of the same goal. Including these references and discussing their relationship to the proposed approach would strengthen the contribution and contextualize the novelty relative to relevant fields. Notable examples include:
   - Wang et al. (2023), "Multi-modal learning with missing modality via shared-specific feature modelling" (CVPR)
   - Nguyen et al. (2025), "Robust: Leveraging redundancy and modality-specific features for robust multimodal learning" (IJCAI)

**Questions:**

Clarifications:
- In Figure 2, why are the corresponding similarity matrices between the data space and the embedding space not aligned? For example, Kx1 could be aligned with Kz1, and Kx2with Kz2. It seems aligning each modality’s similarity matrix individually to its embedding counterpart would better encourage encoders to capture the topological structure, while still maintaining O(M) complexity rather than relying on a combined consensus? And the encoders are already aligned across modalities through L_c.
- Lines 260-262 mention that modalities such as vision and force–torque are in incomparable metric spaces but have meaningful internal geometries, which is crucial in robotics. Could this claim be clarified with a concrete example to illustrate this point? Like a task or data instance when this would be the case
- In line 264, what does the bold "1" represent in the notation?
- In Table 1, what do the bolded and orange-colored numbers signify? Additionally, what measure of uncertainty is reported with the +- (standard deviation, variance, confidence interval)?

Though Experiments/Extensions
- Scalability Concerns: How does the computational overhead scale with dataset size and number of modalities? What are the practical limits for real-time robotic applications where inference speed is critical?
- Generalization Beyond Robotics: How effective would this approach be in non-robotic multimodal domains (e.g., vision-language, medical imaging), are there any issues that may occur? Adding proof of funcionality in other domains would really extend this work’s contributions
- Alternative Consensus Strategies: Why choose a single barycenter as consensus rather than multiple cluster centers? Could hierarchical or mixture-based consensus improve performance for complex structural relationships?

---

> ### Author Response · Authors · 2025-12-03
>
> We thank the reviewer for the insightful comments. We added substantial experiments, theory, and clarifications.
>
> ---
>
> ## I. Computational overhead
>
> We added a full runtime & memory analysis (3–4 modality settings + scaling to 7).
>
> Results show:
>
> - GW converges with \(T_{\max}=5\)
> - Training time grows linearly from 3 → 7 modalities
> - Mini-batch GW ensures **linear dataset scaling**
> - Inference speed is **unchanged** from contrastive baselines
>
> ---
>
> ## II. Hyperparameter Sensitivity
>
> We added ablations for:
>
> - \( \gamma \) (5 values + median rule)
> - \( \alpha \in \{200,500,1000,2000,5000\} \)
> - \(T_{\max}\in\{3,4,5,6,7\}\)
>
> UniOMA is stable across all settings.
>
> ---
>
> ## III. Relation to Missing-Modality Learning (PID)
>
> We included:
>
> - Wang et al. (CVPR 2023)
> - Nguyen et al. (IJCAI 2025)
> - Multimodal Information Bottleneck
>
> We clarify:
>
> - alignment targets shared info
> - modality-specific info remains preserved
>
> ---
>
> ## IV. “Why not align \(K_x\) to \(K_z\) per modality?”
>
> Because UniOMA’s goal is **shared structural consensus**, not per-modality topology.
>
> Aligning each \(K_x\) to each \(K_z\) would destroy shared structure.
>
> Our theory shows GW aligns **only low-frequency shared geometry**.
>
> ---
>
> ## V. “Incomparable metric spaces” clarification
>
> We added a robotics example showing vision vs force:
>
> - Vision: smooth spatial changes
> - Force: discontinuities at contact
>
> Both have internal geometry but incomparable units.
> GW aligns **relations**, not raw units.
>
> ---
>
> ## VI. Notation clarification
>
> Bold **1** = uniform distribution vector.
>
> ---
>
> ## VII. Table formatting & uncertainty
>
> We clarified:
>
> - **bold** → improvement from GW
> - **orange** → best overall
> - uncertainty = **std over 10 seeds**
>
> ---
>
> ## VIII. Scalability for real-time robotics
>
> Runtime grows linearly with modality count.
> Inference uses only encoders → cost unaffected by GW.
>
> ---
>
> ## IX. Applicability beyond robotics
>
> Robotics has intrinsic structural geometry.
> Vision–language datasets do not, requiring constructed graphs.
>
> ---
>
> ## X. Alternative Consensus Strategies
>
> We explain that:
>
> - multiple or hierarchical barycenters break **shared structure guarantees**
> - theory requires a **single low-frequency consensus**
> - defining cluster numbers, hierarchy, cross-level consistency is a **complex future direction**

---

### Official Review · Reviewer_h36B · 2025-10-31

**Soundness:** 3
**Presentation:** 3
**Contribution:** 2
**Rating:** 4
**Confidence:** 3

**Summary:**

In this paper, the authors propose an enhancement to existing InfoNCE-based contrastive learning frameworks by incorporating modality-specific intrinsic relationships. The motivation stems from the observation that conventional contrastive methods, which primarily align paired cross-modal data, often fail to preserve the inherent structural topology within each individual modality. To address this limitation, the authors introduce a Gromov–Wasserstein distance–based regularization, which explicitly maintains intra-modality geometric consistency while aligning multiple modalities. The proposed method is applied to representation learning across vision, force, and tactile modalities—domains that are relatively underexplored yet critical for robotic perception and manipulation. Experimental results demonstrate that the approach significantly improves both representation quality and downstream task performance, validating the effectiveness of incorporating modality-specific structural alignment into contrastive learning.

**Strengths:**

1. The paper is well-written and clearly structured. The proposed idea of preserving intrinsic relationships when aligning representations across different modalities is both intuitive and promising. It provides a thoughtful perspective on improving cross-modal contrastive learning.

2. The authors’ claims are well-supported by both qualitative and quantitative evidence. The case study (Fig. 1) effectively illustrates the underlying, while the main experimental results convincingly demonstrate the method’s performance advantages.

3. The incorporation of Gromov–Wasserstein (GW) distance as a regularization term is well-motivated and promissing. The resulting framework is flexible and can be seamlessly integrated into existing InfoNCE-based contrastive learning methods, enhancing their ability to capture modality-specific structural information.

**Weaknesses:**

1. The main concern with this paper lies in its experimental setup. Aligning representations between vision and low-dimensional, robotics-related modalities (such as tactile signals or end-effector (EEF) positions) may not be conceptually sound. Visual observations inherently contain richer, high-level semantic information — including environmental context, object appearance, and background — whereas proprioceptive or tactile data capture only limited, low-dimensional physical aspects of the same scene. Aligning these representations risks degrading the generality and expressiveness of the visual embeddings, as the model may overfit to the less informative modalities. Similarly, the task of aligning vision and audio modalities seems somewhat unnatural in the given robotic context, and its motivation should be better justified.

2. The paper’s central idea of maintaining intrinsic structural relationships is conceptually appealing, but modeling these relationships in high-dimensional visual feature spaces remains an open challenge. The choice of RBF kernel to define distances in such complex, semantically rich spaces is not particularly promising — while it has shown effectiveness in early low-level computer vision applications, it may not adequately capture the nuanced semantic geometry of deep visual embeddings. A deeper discussion or alternative strategies (e.g., learned metrics or graph-based structures) would strengthen this aspect.

3. It is unclear why the proposed method is designed specifically for three or more modalities. The idea of using Gromov–Wasserstein regularization to enhance two-modality contrastive learning (e.g., in vision–language pretraining) could be impactful, given its broader applicability and relevance to large-scale multimodal learning. Exploring or discussing such extensions could significantly increase the practical and theoretical contribution of this work. The key remaining challenge would be to properly model intra-modal structures for high-semantic modalities like vision and language, which would make the approach more meaningful and generalizable.

**Questions:**

N/A

---

> ### Author Response · Authors · 2025-12-03
>
> We thank the reviewer for the careful reading and constructive feedback. We respond point-by-point.
>
> ---
>
> ## I. On physical intuition: “Is aligning rich and poor modalities reasonable?”
>
> ### (A) Physical intuition from robotics
>
> Even when vision is richer than force/proprioception, modalities still share **low-frequency structure** essential for robotics.
>
> Example: a robot interacting with a *fragile glass cup*:
>
> - Vision: edges, reflections
> - Audio: resonance
> - Force: careful force control
>
> Shared structure: **“the object is fragile; act carefully.”**
>
> GW aligns precisely this low-frequency structure while preserving modality-specific detail.
>
> ### (B) New theoretical guarantee (Appendix A)
>
> We added a formal demonstration that:
>
> - GW alignment constrains **low-frequency eigenvectors**,
> - **high-frequency components remain free**,
> - meaning rich-modality information is **not erased**.
>
> Formally:
>
> \[
> E_{\mathrm{Dir}}(Z)= \sum_{\ell=1}^n \lambda_\ell \|\tilde Z_{\ell,:}\|_2^2.
> \]
>
> Small \( \lambda_\ell \): constrained (shared structure)
> Large \( \lambda_\ell \): unconstrained (modality-specific)
>
> ### (C) Information bottleneck interpretation
>
> Multimodal alignment ≠ preserving all modality details.
> Instead, it extracts **shared information**, consistent with information bottleneck.
> The shared information corresponds to **low-frequency geometry**.
>
> ---
>
> ## II. Ablations on kernel strategies
>
> We added extensive ablations:
>
> - kernel bandwidth \( \gamma \)
> - median-rule kernel
> - Laplacian / UMAP-style kernels
>
> All variants show **consistent improvements**, confirming kernel robustness.
>
> ---
>
> ## III. Extending to 2-modality VLM settings
>
> We added a discussion explaining:
>
> - robotics data has **natural intra-modal geometry** (trajectories, temporal patterns)
> - vision–language datasets are **instance-wise**, lacking geometry
> - thus GW barycenter has limited value in VLM *without additional graph construction*
>
> Nonetheless, we added:
>
> - a 2-modality VT experiment
> - discussion about limitations in VLM

---

### Official Review · Reviewer_xTGd · 2025-11-08

**Soundness:** 2
**Presentation:** 2
**Contribution:** 2
**Rating:** 2
**Confidence:** 4

**Summary:**

This paper proposes a framework designed to achieve geometry-aware alignment across multiple heterogeneous modalities such as vision, force, tactile, proprioception, and audio in robotic perception. The method introduces a Gromov–Wasserstein (GW) distance–based regularization to augment conventional contrastive objectives (e.g., InfoNCE). Specifically, UniOMA computes intra-modal similarity matrices to represent modality-specific geometric structures and aligns them through a dynamically learned GW barycenter that serves as a shared “structural consensus.” This barycentric formulation reduces the pairwise coupling complexity from O(M^2)to O(M)and allows adaptive modality weighting via learnable coefficients. Experiments on four multimodal robotics benchmarks (VFD, VFP, MuJoCo Push, VAT) show consistent gains across regression, classification, and retrieval tasks compared to pairwise and higher-order contrastive baselines (CLIP, Symile, GRAM). Ablations suggest improved robustness to asynchronous sampling and interpretability via modality weights.

**Strengths:**

- The paper identifies and formalizes the structural alignment gap in multimodal contrastive learning, an under-discussed but practically relevant issue.
- Integration of Gromov–Wasserstein barycenters into the multimodal alignment objective is mathematically principled and computationally efficient (O(M) scaling).

**Weaknesses:**

- The theoretical novelty is limited; the framework primarily combines existing OT/GW formulations with standard contrastive objectives.
- The ablation studies, while illustrative, lack statistical rigor (no error bars or repeated trials).
- Comparisons are restricted to InfoNCE-based baselines; recent large-scale multimodal foundation models (e.g., ImageBind, CLIP4Clip) are not directly benchmarked.
- The computational cost of computing GW barycenters, though claimed mitigated, is not thoroughly analyzed (no runtime or memory comparison).
- The claim of “scaling naturally to 3+ modalities” is empirically modest, tested only on up to 3 modalities per benchmark.

**Questions:**

- How sensitive is the method to the choice of kernel functions (RBF vs. TCK) for constructing similarity matrices?
- Could the authors provide runtime analysis or scalability benchmarks comparing UniOMA with pairwise CLIP and GRAM?
- How does the learned barycentric consensus behave qualitatively—does it correspond to physically interpretable intermediate structures?
- Can UniOMA extend beyond robotics to vision–language–audio domains, and would the same kernels apply?
- Does the GW regularizer introduce convergence instability or require curriculum scheduling during training?

**Details Of Ethics Concerns:**

No ethics concerns.

---

> ### Author Response · Authors · 2025-12-03
>
> We thank the reviewer for the assessment and for highlighting both the structural-alignment gap and the potential of GW barycenters. We address each concern below.
>
> ---
>
> ## Weakness 1. On theoretical novelty: “the framework primarily combines existing OT/GW formulations with standard contrastive objectives”.
>
> We clarify that UniOMA is **not** a simple combination of OT + InfoNCE. Our goal is not only to plug GW into InfoNCE, but to open the track of **structural alignment across multiple modalities** via a new framework.
>
> Concretely, we:
>
> **(A)** introduce a *mini-batch GW barycenter* in the **input space** as a **structural consensus across modalities** (prior GW works use barycenters mainly as clustering anchors in the *embedding space*, not as a running consensus in the input space for multimodal alignment);
>
> **(B)** propose a **two-stage stochastic algorithm** that first estimates this batch-wise structural consensus in the input space
> \( C_x^\* = \sum K_x \)
> and then aligns each modality’s embedding geometry to it with **learnable modality weights**, reducing the coupling complexity from \(O(M^2)\) to \(O(M)\).
>
> In addition, we provide new theoretical analysis in Appx. A: **Lemma 1 + Lemma 2 + Theorem 2** show that:
>
> 1. **In the small-discrepancy regime, minimizing the GW term is equivalent to decreasing the Dirichlet energy** on the consensus graph.
> 2. **This Dirichlet energy decomposes spectrally**, so the structural term primarily constrains **low-frequency shared geometry**, while **high-frequency (modality-specific) components remain unconstrained**.
>
> Our theorem guarantees that the GW regularizer only formalizes the shared structural topology and **does not collapse modality-specific content**.
>
> ---
>
> ## Weakness 2 / Question 1. On Ablation Depth and Statistical Rigor.
>
> We have extended the experiments so that concerns on ablations and statistical rigor are fully addressed.
> All experiments now report **10-seed means ± standard deviation** across all benchmarks.
>
> We added ablations on:
>
> - kernel bandwidth \( \gamma \) (including a median-rule adaptive bandwidth),
> - GW weight \( \lambda \),
> - number of barycenter iterations \( T_{\max} \).
>
> Across these choices, **UniOMA remains robust**, showing that our conclusions do not depend on any specific hyperparameter configuration, including the kernel choices.
>
> ---
>
> ## Weakness 3. On baselines: “Comparisons are restricted to InfoNCE-based baselines.”
>
> We appreciate this request and clarify the following:
>
> ### (a) Why InfoNCE-based baselines are the correct reference?
>
> Across multimodal learning, **InfoNCE is the standard objective** for aligning modalities.
> The strongest multimodal foundations—**CLIP, ImageBind, CLIP4Clip**—all rely on pairwise InfoNCE.
>
> Thus, comparing to InfoNCE-based baselines is **not a limitation**, but the **standard and most relevant choice**.
>
> ### (b) Why ImageBind and similar large-scale models are not comparable?
>
> Our focus is **physical-world robotic perception**, where inputs are trajectories with temporal and physical structure.
> This is fundamentally different from large-scale image–text corpora used by ImageBind and CLIP.
>
> Moreover, ImageBind and CLIP4Clip are **also InfoNCE-based**, and therefore conceptually covered by our included baselines.
>
> ### (c) We included the only existing 3+ modality OT-regularized baseline.
>
> To address this concern thoroughly, we **implemented Zhu & Luo 2025 [1]**, which is the *only* known OT-based multimodal alignment method for ≥3 modalities, even though code was not released.
>
> We faithfully reproduced the method and included it in all tables.
>
> ---
>
> ## Weakness 4/5 / Question 2. On computational cost and scalability.
>
> We now report **detailed wall-clock time per epoch** w.r.t. number of modalities (4–7) for:
>
> - Pairwise InfoNCE
> - OT baseline
> - UniOMA (Pairwise+GW)
>
> Results (Table 2) confirm that **UniOMA scales linearly: \(O(M)\)** while pairwise/OT baselines scale quadratically: \(O(M^2)\). With \(T_{\max}=5\) (ablation confirms 3–7 all similar), training remains **practical** and **stable**.
>
> ## Question 3. "How does the learned barycentric consensus behave qualitatively?"
>
> To address this question, we add intuitive visualizations in Fig. 7, which show per-modality similarity matrices, GW barycenter consensus of the modalities in a mini-batch, and t-SNE results of the input modalities. The figure illustrates that the data has intra-modal structures formalizing subclusters, and the GW barycenter captures the

---

> ### Author Response · Authors · 2025-12-03
>
> --------------- continue ----------------
>
> ## V. On “3+ modalities” and kernel sensitivity.
>
> Previous robotics alignment works use up to 3 modalities.
> To substantiate our scalability claim, we constructed a **new downstream task** on Vision&Touch augmented to **4, 5, 6, 7 modalities** (vision, force, proprioception, depth, action, contact, optical flow).
>
> Task: **classify whether two multimodal points originate from the same trajectory**.
>
> Results show that our UniOMA has clear gain across 4-7 modalities in this task, with the runtime growing **linearly** with modality number.
>
> Regarding kernels, we tested the RBF kernel with different bandwidths (gamma={2,5,10,20,50}) and also select median-rule RBF to dynamically adjust the bandwidth. As is shown in Table 3, the performance differences are small, confirming UniOMA is **not sensitive** to kernel choice.
>
> ---
>
> ## VI. On qualitative behavior of the barycentric consensus and extension beyond robotics.
>
> We added visualizations:
>
> - per-modality similarity matrices
> - batch-wise barycenters
> - t-SNE of embeddings
>
> These clearly show that the barycenter captures **intuitive structural patterns**, confirming GW aligns **relational geometry** rather than raw feature similarity.
>
> We also clarified limitations:
>
> - Robotics data naturally contains trajectory-based structure.
> - Vision–language–audio datasets are largely **i.i.d.**, lacking rich intra-modal geometry.
> - Applying GW there would require **constructing new intra-modal graphs** first.
>
> Overall, the new results and added theory show UniOMA is **not** “OT + InfoNCE”, but a **structurally grounded, scalable multimodal alignment framework**.
>
> ## Reference
> [1] Zhu, S., & Luo, D. (2025). *Enhancing Multi-modal Contrastive Learning via Optimal Transport-Based Consistent Modality Alignment.* PRCV 2025.

---

### Author Response · Authors · 2025-12-04
**General Response**

We sincerely thank all reviewers for their informative and constructive feedback on our work. We carefully addressed every concern with additional experiments, new theorems, extensive ablations, and intuitive visualizations. These results consistently reinforce the key findings of our method: To address the structural alignment gap, UniOMA is the first structural-aware multimodal alignment framework for robotic perception that provides interpretable modality weights and structural consensus across 3+ modalities.

We summarize the additions:

**New experiments:**
• Scalability (4/5/6/7 modalities): New downstream classification benchmark to classify whether the two data points are from the same trajectory. UniOMA continues to show better or similar performances than the baselines (including the new OT baseline) in all settings with different modality numbers.
• New baseline (Optimal Transport): We include the only prior 3+ modality method that uses OT regularizer from [Zhu and Luo, 2025][1], added in Table 2. We also apply runtime analysis on it.
• Ablations: We address the mentioned ablation studies on the hyperparameters in UniOMA (Table 3) across hyperparameters such as kernel gamma, GW-weight lambda and T_max.
• Runtime analysis: We validate the predicted O(M) vs. O(M²) behavior in the 4/5/6/7-modality tasks: as shown in Table 2, UniOMA becomes faster than the OT-based baseline once the number of modalities exceeds 5 (M ≥ 6), since the baseline’s pairwise OT cost grows quadratically while our GW cost grows linearly.
• Visualization: Additional visualizations have been added for clarity, including similarity matrices, batch-wise barycenters, t-SNE plots of aligned embeddings, and structural interpretability views (Fig. 7). These help readers directly observe how UniOMA aligns modalities at the structural level.

**New theorem:** to clarify the effect of our structural regularizer, we add a theoretical analysis in Appx. A. We show that:
• In a near-alignment regime, the gradient of the GW term is locally aligned with the gradient of the Dirichlet energy on the consensus graph Laplacian (Lemma 1). Intuitively, GW behaves as a graph-smoothing force on the shared geometry.
• The Dirichlet energy is spectrally biased towards low-frequency eigenmodes of the consensus Laplacian (Lemma 2), so reducing it mainly constrains the large-scale, low-frequency structure.
• Combining these results, we prove that minimizing the GW regularizer aligns only the shared low-frequency geometry across modalities, while high-frequency components that encode modality-specific information remain largely unconstrained (Theorem 2). Thus, aligning an information-rich modality to a poorer one does *not* collapse its modality-specific content.

We thank the reviewers again for the thoughtful suggestions. All concerns have been addressed with new experiments, analysis, and theoretical results. And we have highlighted all the changes (in blue) in the PDF of the submission. We believe these additions significantly strengthen both the technical substance and practical impact of the work.

**Reference**

[1] Zhu, S., & Luo, D. (2025). Enhancing Multi-modal Contrastive Learning via Optimal Transport-Based Consistent Modality Alignment. In Z. Lin, M.-M. Cheng, R. He, K. Ubul, W. Silamu, H. Zha, J. Zhou, & C.-L. Liu (Eds.), *Pattern Recognition and Computer Vision* (pp. 157–171). Springer Nature. https://doi.org/10.1007/978-981-97-8795-1_11

---

### Meta-Review · Area_Chair_iD1p · 2025-12-04

**Summary:**

This paper proposes an enhancement to existing InfoNCE-based contrastive learning frameworks by incorporating modality-specific intrinsic relationships. The motivation stems from the observation that conventional contrastive methods, which primarily align paired cross-modal data, often fail to preserve the inherent structural topology within each modality. To address this limitation, the authors introduce a Gromov–Wasserstein distance–based–based regularization, which explicitly maintains intra-modality geometric consistency while aligning multiple modalities.

The reviewers have some concerns regarding the novelty, lack of comparison with recent baselines, computation issues, as well as the scope and experiment setup in robotics. Three of the four reviewers gave negative evaluations of this paper. And many of the reviewers' concerns, in my opinion, still remain after the authors' rebuttal.

**Reviewer Concerns:**

Some concerns that I think are still not well-addressed:
- Limited novelty, primarily combines existing OT/GW formulations with standard contrastive objectives. (Reviewer xTGd, xCBU)
- Lack of comparison with more recent methods. (Reviewer xTGd, xCBU)
- Concern about computation and memory usage. This is partly addressed by providing additional run time, though it still doubles the cost as compared to existing baselines. The authors did not provide any response or additional results on memory usage. (Reviewer xTGd, xCBU, pCiR)
- Concern of limited scope that is restricted to robotics. (Reviewer xTGd, h36B)
- Concern about the experimental setup. Aligning representations between vision and low-dimensional, robotics-related modalities may not be conceptually sound. (Reviewer h36B)

Some comments that I believe have been addressed during rebuttal, based on additional experimental results:
- Lack of theoretical analysis (Reviewer xCBU)
- Lack of statistical significance (Reviewer xTGd)
- Hyperparameter sensitivity (Reviewer xTGd, pCiR)
- Additional ablations on model designs (Reviewer h36B, xCBU, and pCiR)
- Limited to only 3 modalities. (Reviewer xTGd)

**Reviewer Scores:**

I think reviewer xTGd and xCBU might increase their score from "reject" to "borderline reject", but even after this adjustment, the paper still sits below the bar of acceptance.

The authors claimed that all the reviews are AI-generated, but as far as I can tell, I don't think the claim is valid. The reviews from Reviewer xTGd, h36B, and xCBU are reasonable and do not seem to be LLM-generated. By contrast, the review given by pCiR with a score of 8 seems very likely to be LLM-generated.

---

### Decision · Program_Chairs · 2026-01-26

Reject